



# fenics_ice 1.0: A framework for quantifying initialisation uncertainty for time-dependent ice-sheet models

Conrad Koziol[1], Joe Todd[1], Daniel Goldberg[1], and James Maddison[2]

[1]School of GeoSciences, Univ. of Edinburgh, City of Edinburgh, United Kingdom
[2]School of Mathematics, Univ. of Edinburgh, City of Edinburgh, United Kingdom

**Correspondence:** D N Goldberg (dan.goldberg@ed.ac.uk)

**Abstract.**

Mass loss due to dynamic changes in ice sheets is a significant contributor to sea level rise, and this contribution is expected to increase in the future. Numerical codes simulating the evolution of ice sheets can potentially quantify this future contribution. However, the uncertainty inherent in these models propagates into projections of sea level rise, and hence is crucial to understand. Key variables of ice sheet models, such as basal drag or ice stiffness, are typically initialized using inversion methodologies to ensure that models match present observations. Such inversions often involve tens or hundreds of thousands of parameters, with unknown uncertainties and dependencies. The computationally intensive nature of inversions along with their high number of parameters mean traditional methods such as Monte Carlo are expensive for uncertainty quantification. Here we develop a framework to estimate the posterior uncertainty of inversions, and project them onto sea level change projections over the decadal timescale. The framework treats parametric uncertainty as multivariate Gaussian, and exploits the equivalence between the Hessian of the model and the inverse covariance of the parameter set. The former is computed efficiently via algorithmic differentiation, and the posterior covariance is propagated in time using a time-dependent model adjoint to produce projection error bars. This work represents an important step in quantifying the internal uncertainty of projections of ice-sheet models.

## 1 Introduction

The dynamics of ice sheets are strongly controlled by a number of physical properties which are difficult (or intractable) to observe directly, such as basal traction and ice stiffness (Arthern et al., 2015). This poses challenges in terms of future ice-sheet projections, as the behaviour of ice sheets often depends strongly on these (spatially varying) properties. There are two principal approaches that have been taken by ice-sheet modellers to approach these challenges: control methods and sampling-based uncertainty quantification. Below, we discuss these approaches in the context of ice-sheet modelling.

Control methods (MacAyeal, 1992), sometimes referred to simply as "inverse methods" in a glacial flow-modelling context, consist of the minimisation of a "cost" function involving some global measure of model-data misfit, as well as regularisation cost terms which penalise nonphysical behaviour (e.g. high variability at small scales or strong deviation from prior knowledge). A strong benefit of control methods is their ability to estimate hidden properties at the grid scale through large-scale





optimisation techniques. Such methods have been used extensively to calibrate ice-sheet models to observations (e.g., Rommelaere, 1997; Vieli and Payne, 2003; Larour et al., 2005; Sergienko et al., 2008; Morlighem et al., 2010; Joughin et al., 2010; Fürst et al., 2015; Cornford et al., 2015).

Uncertainty quantification (UQ) in projections of ice-sheet behaviour is a crucial challenge in ice-sheet modelling. Studies of fast-flowing Antarctic glaciers have shown that uncertainties in the parameters controlling ice flow can lead to large variability

in modelled behaviour (Nias et al., 2016). Thus it is of great importance to quantify how this parametric uncertainty translates into uncertainty in projections. In some cases, this uncertainty may be exogenous to the dynamics of the ice sheet model: for instance, uncertainty in ocean-driven ice shelf melt, while a likely important contributor to ice-sheet projection uncertainty (Robel et al., 2019), arises from incomplete knowledge of the ocean system rather than the dynamics of the ice model itself. This is in contrast to parameters that must be constrained via calibration; their uncertainties derive from observational uncertainty,

uncertainty in model physics, and *a priori* knowledge.

The uncertainty associated with ice-sheet model calibration can be quantified through Bayesian inference, in which prior knowledge is "updated" with observational evidence. Such methods have been applied to continental-scale ice-sheet models and models of coupled ice-ocean interactions (Gladstone et al., 2012; Ritz et al., 2015; Deconto and Pollard, 2016). In these Bayesian studies, the dimension of the parameter space is small (i.e. less than $\sim$20). Though the methods of these studies differ,

they share the common feature of generation of a large ensemble (thousands of runs) through sampling of a parameter space. Bayesian methods are then applied in conjunction with observational data to find likelihood information for the parameters, and associated probability distributions of ice-sheet behaviour.

Applying such ensemble-based Bayesian methods to glacial flow models and parameter sets of dimension $\sim \mathcal{O}(10^4 - 10^6)$ (a dimension size typical of control methods) is prohibited by computational expense. Although control methods efficiently

provide estimates of parameter fields through gradient-based optimisation, they do not provide parametric uncertainty. While it can be shown that such methods provide the *most likely* parameter field (often referred to as the Maximum A Posteriori, or *MAP*, estimate) (Raymond and Gudmundsson, 2009; Isaac et al., 2015), the *covariance* of the joint probability distribution – necessary for assessing uncertainty in calibrated model behaviour at the MAP point – cannot be inferred.

Thus, there is at present a disconnect between the dual aims of (i) modelling ice sheets as realistically as possible, i.e. through

the implementation of higher-order stresses and without making limiting assumptions regarding "hidden" properties of the ice sheet, and (ii) uncertainty quantification (UQ) of models by approximate inference by reducing the dimensionality of the set of parameters.

By augmenting control methods using a Hessian-based Bayesian approach, it is possible to quantify parametric uncertainty without sacrificing parameter dimension or model fidelity. Just as control methods can be interpreted as returning the *mode*

of a joint posterior probability distribution, it can be shown that, under certain assumptions, the *covariance* of the distribution can be characterised by the inverse of the *Hessian* (the matrix of second derivatives) of the cost function with respect to the parameters (Thacker, 1989; Kalmikov and Heimbach, 2014; Isaac et al., 2015). For a nonlinear model, calculating the Hessian involves model second derivatives with respect to parameters, which can be challenging for complex models; in many cases, second-derivative information is ignored and the Hessian is approximated using first-derivative information only (Kaminski





et al., 2015); such an approximation is referred to as the Gauss-Newton Hessian (Chen, 2011). Some studies retain second-derivative information, however, using variational methods (Isaac et al., 2015) or Algorithmic Differentiation (AD) software (Kalmikov and Heimbach, 2014).

Once determined, the Hessian-based parameter covariance can then be used to quantify the variance of a scalar Quantity of Interest (QoI) of the calibrated model (e.g., ice-sheet sea level contribution over a specified period). One approach to this is projecting the parameter covariance on to a linearised model prediction (e.g., Kalmikov and Heimbach, 2014). Isaac et al. (2015) employ this methodology in a finite-element ice flow model, but since their model is time-independent, uncertainty estimates cannot be projected forward in time.

In this study we introduce a framework for time-dependent ice-sheet uncertainty quantification, and apply it to an idealised ice-sheet flow problem (Pattyn et al., 2008). Beginning with a cost-function optimisation for sliding parameters given noisy ice-sheet velocity data, we then generate a low-rank approximation to the posterior covariance of the sliding parameters through the use of the cost-function Hessian. In our work, the Hessian is calculated through AD, using the "complete" Hessian rather than the Gauss-Newton approximation. We then project the covariance on a linearisation of the time-dependent ice-sheet model (again using AD to generate the linearisation) to estimate the growth of QoI uncertainty over time. We also apply a method of sampling the posterior distribution, and use this to validate our calculation of time-dependent QoI uncertainty for an idealised problem.

## 2 Methodology

### 2.1 Symbolic convention

To facilitate readability of this and subsequent sections we adopt formatting conventions for different mathematical objects. Coefficient vectors corresponding to finite-element functions appear as $\bar{c}$; general vectors and vector-valued functions as $\breve{d} \in \mathbb{R}^q$; and matrices as $\mathbf{E}$.

### 2.2 Mathematical Framework

An ice-sheet flow model can be thought of as a (nonlinear) mapping from a set of input fields, which might be unobservable or poorly known (such as bed friction) to a set of output fields, which might correspond to observable quantities (such as surface velocity). Here, our focus is on the probability distribution function (PDF) of a "hidden" field $C$ *conditioned* on an observational field $U$, i.e. $p(C|U)$; and our aim is to determine properties of this conditional distribution through Bayes' theorem:

$$p(C|U) = \frac{p(U|C)p(C)}{p(U)}. \tag{1}$$

$p(U)$, the unconditional distribution of observations, is effectively a normalisation constant which we do not consider further.

As described in Section 3, our ice-sheet flow model is a finite-element model, meaning $C$ can be described by a vector of finite dimension. We furthermore consider discrete observations, meaning $U$ can be described by a finite-dimensional vector





as well (in general with different dimension from $C$). We assume that observational errors follow a Gaussian distribution. Referring to the vector of observations as $\breve{u}_{obs} \in \mathbb{R}^m$, this is expressed as

$$-\log\left(p(\breve{u}_{obs})\right) = \frac{1}{2}\langle \breve{u}_{obs} - \breve{u}_{true}, \breve{u}_{obs} - \breve{u}_{true}\rangle_{\boldsymbol{\Gamma}_{obs}^{-1}} \equiv \frac{1}{2}\|\breve{u}_{obs} - \breve{u}_{true}\|_{\boldsymbol{\Gamma}_{obs}^{-1}}^2. \tag{2}$$

Here, $\langle \breve{a}, \breve{b}\rangle_{\boldsymbol{\Gamma}_{obs}^{-1}}$ is the Euclidean inner product of $\breve{a}$ with $\boldsymbol{\Gamma}_{obs}\breve{b}$, where $\boldsymbol{\Gamma}_{obs} \in \mathrm{Sym}^+(m)$ (the set of real symmetric positive definite $m \times m$ matrices) is the observational *covariance matrix*, and $\|\breve{a}\|_{\boldsymbol{\Gamma}_{obs}}$ is the associated norm. If the parameter field is represented by the vector $\overline{c} \in \mathbb{R}^n$, then the conditional PDF $p(U|C)$ satisfies

$$-\log\left(p(\breve{u}_{obs}|\overline{c})\right) = \frac{1}{2}\|\breve{u}_{obs} - \breve{f}(\overline{c})\|_{\boldsymbol{\Gamma}_{obs}^{-1}}^2, \tag{3}$$

where $\breve{f}: \mathbb{R}^n \to \mathbb{R}^m$ is a function from the space of parameter fields to the space of observations, i.e. our ice-sheet flow model. Note that the above construction equates $\breve{f}(\overline{c})$ with the "truth", i.e. it assumes zero model error. In general model error is extremely difficult to constrain, and doing so is beyond the scope of our study; however, in Section 6 we discuss potential strategies to incorporate model error into our framework.

The distribution $p(C)$ in Eq. 1 is the *prior* PDF of $\overline{c}$, which expresses knowledge of $C$ prior to consideration of ice-sheet observations and physics – for instance, the autocorrelation scale of basal friction, which may be inferred from proxies such as the presence of basal water inferred from ice-penetrating radar. If the prior PDF is Gaussian, then the distribution of $\overline{c}$ conditioned on $\breve{u}_{obs}$ satisfies

$$-\log\left(p(\overline{c}|\breve{u}_{obs})\right) = \frac{1}{2}\|\breve{u}_{obs} - \breve{f}(\overline{c})\|_{\boldsymbol{\Gamma}_{obs}^{-1}}^2 + \frac{1}{2}\|\overline{c} - \overline{c}_0\|_{\boldsymbol{\Gamma}_{prior}^{-1}}^2, \tag{4}$$

where $\overline{c}_0$ is the prior mean and $\boldsymbol{\Gamma}_{prior} \in \mathrm{Sym}^+(n)$ is the prior covariance. This conditional distribution is referred to as the *posterior* distribution, or $p_{post}$. If $\breve{f}$ is linear, $p_{post}$ is Gaussian, with mean $\mu$ and covariance $\boldsymbol{\Gamma}$ given by

$$\mu_{post,lin} = \boldsymbol{\Gamma}_{post,lin}\left(\left(\frac{\partial \breve{f}}{\partial \overline{c}}\right)^T \boldsymbol{\Gamma}_{obs}^{-1}(\breve{u}_{obs} - \breve{f}_0) + \boldsymbol{\Gamma}_{post,lin}^{-1}\overline{c}_0\right),$$

$$\boldsymbol{\Gamma}_{post,lin} = \left(\left(\frac{\partial \breve{f}}{\partial \overline{c}}\right)^T \boldsymbol{\Gamma}_{obs}^{-1}\left(\frac{\partial \breve{f}}{\partial \overline{c}}\right) + \boldsymbol{\Gamma}_{prior}^{-1}\right)^{-1}. \tag{5}$$

(The above can be derived by maximizing Eq. 4 with $\breve{f} = \breve{f}_0 + (\partial \breve{f}/\partial \overline{c}_0)(\overline{c} - \overline{c}_0)$.)

Models of ice-sheet dynamics are in general nonlinear, however, and Eq . 5 does not strictly apply. Instead we use a quadratic approximation to the negative log posterior (Bui-Thanh et al., 2013; Isaac et al., 2015; Kalmikov and Heimbach, 2014). Such an approximation considers a second-order Taylor expansion of $-\log(p_{post})$ about the mode of the posterior, or equivalently about the Maximum a Posteriori (MAP) estimate $\overline{c}_{MAP}$. This leads to a Gaussian distribution with mean $\overline{c}_{MAP}$ and covariance

$$\boldsymbol{\Gamma}_{post} = \left(\left(\frac{\partial \breve{f}}{\partial \overline{c}}\right)^T \boldsymbol{\Gamma}_{obs}^{-1}\left(\frac{\partial \breve{f}}{\partial \overline{c}}\right) + \boldsymbol{\Gamma}_{prior}^{-1} + \left(\frac{\partial^2 \breve{f}}{\partial \overline{c}^2}\right)\boldsymbol{\Gamma}_{obs}^{-1}(\breve{u}_{obs} - \breve{f}(\overline{c}))\right)^{-1}. \tag{6}$$





Eq. 6 differs from the covariance given by Eq. 5 in that derivatives of $\breve{f}$ depend on $\bar{c}_{MAP}$, and in the final term involving second derivatives of $\breve{f}$. Essentially, $p_{post}$ is approximated by the Gaussian distribution with the local covariance at $\bar{c}_{MAP}$. While this is insufficient to calculate global properties of $p_{post}$ such as skew, it gives insight into the directions in parameter space which are most (and least) constrained – information which can be propagated to model projections.

## 2.3 Relation to control methods

By contrast with Bayesian methods, the control methods discussed in Section 1 find the parameter set which gives the best fit to observations. This is done by minimizing a scalar cost function which takes the general form

$$J^c = J^c_{mis} + J^c_{reg}. \tag{7}$$

$J^c_{mis}$, the misfit cost, is the square-integral of the misfit between the surface velocity of the ice model and remotely-sensed observations, normalised by the observational error. These terms are discretised to implement the control method. If the ice-sheet model is solved via a finite element scheme, then the misfit cost can be written

$$J^c_{mis} = \frac{1}{2}\|\breve{u}_{obs} - \breve{u}\|^2_{\mathbf{D}_\sigma{}^{-1}\mathbf{M}\mathbf{D}_\sigma{}^{-1}} \tag{8}$$

Here $\breve{u}$ and $\breve{u}_{obs}$ are nodal values of the finite-element representations of modelled and observed velocities; $\mathbf{D}_\sigma$ is a diagonal matrix containing standard errors of the $\breve{u}_{obs}$ measurements; and $\mathbf{M}$ is the *mass matrix* corresponding to the finite element basis $\phi_i$: $M_{ij} = \int_\Omega \phi_i \phi_j dA$, where $\Omega$ is the computational domain. $J_{reg}$, the regularisation cost, is imposed to prevent instabilities, and is generally chosen as a Tikhonov operator which penalises the square-integral of the gradient of the parameter field (e.g., Morlighem et al., 2010; Cornford et al., 2015). In other words, regularisation imposes smoothness on the control parameter field, which otherwise may exhibit variability at scales not strongly determined by the observations. Such a term can generally be written as a positive definite quadratic form of $\bar{c}$.

$J^c$ is thus a functional with a form similar to Eq. 4, i.e. the negative log posterior. In this sense, solving the control problem is equivalent to finding $\bar{c}_{MAP}$. However, there are important differences between $J^c_{mis}$ and the first term of Eq. 4. The former is an $L_2$ inner product (which, with standard continuous finite elements, introduces mesh dependent factors in the covariance) while the latter is an inner product involving values at a fixed set of observation points (which does not). Identifying $J_c$ as a negative log posterior therefore implies observational errors that are changed by factors related to grid cell areas.

Our framework effectively uses a control method – but one which allows calculation of the posterior covariance after the MAP point is found. As such we use a fixed set of points, as described above, in our misfit cost term. Thus, the Hessian of the cost function of our control method is equal to the inverse of the posterior covariance given by Eq. 6. However, our form of $J^c_{reg}$ does not involve the square integral of the gradient of $\bar{c}$, as Bui-Thanh et al. (2013) note this can lead to unbounded prior covariances as the numerical grid is refined. These authors recommend a discretization of a differential operator of the form

$$\mathcal{L}(\cdot) \equiv \gamma \nabla^2(\cdot) - \delta(\cdot) \tag{9}$$





where the second term on the right hand side ensures the operator is invertible. Isaac et al. (2015) use the same definition for their prior, which we adopt in our study as well. Hence, our regularisation term is

$$J_{reg}^c = \int_\Omega \frac{1}{2}(\mathcal{L}(c))^2 dA = \frac{1}{2}\|c\|^2_{\mathbf{LM}^{-1}\mathbf{L}} \tag{10}$$

where $\mathbf{L}$ is the operator on the finite element space such that $\overline{\phi}_i^T \mathbf{L} \overline{\phi}_j = \int_\Omega \mathcal{L}(\phi_i)\mathcal{L}(\phi_j)dA$ for all $\phi_i$, $\phi_j$. Thus, in the Bayesian interpretation of the control method optimisation, the prior covariance is given by

$$\mathbf{\Gamma}_{prior} = \mathbf{L}^{-1}\mathbf{M}\mathbf{L}^{-1}. \tag{11}$$

## 2.4 Low rank approximation

In the previous section we establish that the posterior covariance is equivalent to the inverse of the Hessian of the (suitably defined) cost function. With a large parameter space, though, calculating the complete Hessian (and its inverse) can become computationally intractable. Still, in many cases, the constraints on parameter space provided by observations can be described by a subspace of lower dimension. In the present study, our idealised examples are small enough that the full Hessian can be calculated; but to provide scalable code we seek an approximation to the posterior covariance that exploits this low-rank structure.

The following low-rank approximation follows from Isaac et al. (2015). We define the term

$$\left(\frac{\partial \breve{f}}{\partial \overline{c}}\right)^T \mathbf{\Gamma}_{obs}^{-1}\left(\frac{\partial \breve{f}}{\partial \overline{c}}\right) + \left(\frac{\partial^2 \breve{f}}{\partial \overline{c}^2}\right)\mathbf{\Gamma}_{obs}^{-1}(\breve{u}_{obs} - \breve{f}(\overline{c}))$$

from Eq. 6 as $\mathbf{H_{mis}}$, the Hessian of the misfit component of the negative log posterior (or, equivalently, of the misfit cost term). Eq. 6 can be written

$$\mathbf{\Gamma}_{post} = \left(\mathbf{H}_{mis} + \mathbf{\Gamma}_{prior}^{-1}\right)^{-1}. \tag{12}$$

This can be rearranged:

$$\mathbf{\Gamma}_{post} = \left(\mathbf{\Gamma}_{prior}\mathbf{H}_{mis} + \mathbf{I}\right)^{-1}\mathbf{\Gamma}_{prior}. \tag{13}$$

The term $\tilde{\mathbf{H}}_{mis} \equiv \mathbf{\Gamma}_{prior}\mathbf{H}_{mis}$ is referred to as the "prior-preconditioned Hessian", and it has the eigendecomposition

$$\tilde{\mathbf{H}}_{mis} = \mathbf{C}\mathbf{\Lambda}\mathbf{C}^{-1} \tag{14}$$

where $\mathbf{\Lambda}$ is a diagonal matrix of eigenvalues and $\mathbf{C}$ contains the corresponding eigenvectors. $\tilde{\mathbf{H}}_{mis}$ is not in general symmetric positive semidefinite (even though $\mathbf{H}_{mis}$ and $\mathbf{\Gamma}_{prior}$ both are), but Eq. 14 can be written as

$$\mathbf{\Gamma}_{prior}^{\frac{1}{2}}\mathbf{H}_{mis}\mathbf{\Gamma}_{prior}^{\frac{1}{2}} = \mathbf{\Gamma}_{prior}^{-\frac{1}{2}}\mathbf{C}\mathbf{\Lambda}\mathbf{C}^{-1}\mathbf{\Gamma}_{prior}^{\frac{1}{2}} \tag{15}$$





i.e. an eigendecomposition of the symmetric matrix $\Gamma_{prior}^{\frac{1}{2}}\mathbf{H}_{mis}\Gamma_{prior}^{\frac{1}{2}}$. Thus the eigenvalues in $\Lambda$ are real-valued, and the eigenvectors $\mathbf{C}$ can be chosen to be $\Gamma_{prior}^{-1}$-orthogonal, i.e. such that

$$\mathbf{C}^{T}\Gamma_{prior}^{-1}\mathbf{C} = \mathbf{I}. \tag{16}$$

While $\mathbf{H}_{mis}$ could be eigendecomposed directly, decomposing $\tilde{\mathbf{H}}_{mis}$ better informs uncertainty quantification: for an eigen-vector $\bar{c}_k$ with eigenvalue $\lambda_k$, the negative log posterior probability density evaluated at $\bar{c} = \bar{c}_k + \bar{c}_{MAP}$ is

$$
\begin{aligned}
\langle \bar{c}_k, \mathbf{H}\bar{c}_k \rangle &= \langle \bar{c}_k, (\mathbf{H}_{mis} + \Gamma_{\mathbf{prior}}^{-1})\bar{c}_k \rangle \\
&= (1 + \lambda_k)\langle \bar{c}_k, \Gamma_{\mathbf{prior}}^{-1}\bar{c}_k \rangle \\
&= (1 + \lambda_k).
\end{aligned}
\tag{17}
$$

In other words, the leading eigenmodes of $\tilde{\mathbf{H}}_{mis}$ correspond to those directions in which the posterior uncertainty is reduced by the most, relative to the prior uncertainty in those directions. Thus one can truncate the eigendecomposition, neglecting eigenmodes for which the data provides minimal information. The Sherman–Morrison–Woodbury matrix inversion lemma gives

$$\Gamma_{post} = (\mathbf{I} - \mathbf{CDC}^{-1})\Gamma_{prior} \tag{18}$$

where $\mathbf{D}$ is a diagonal matrix with entries $d_{kk} = \lambda_k/(1 + \lambda_k)$, and with Eq. 16 this becomes

$$\Gamma_{post} = \Gamma_{prior} - \mathbf{CDC}^{T}. \tag{19}$$

This can then be approximated

$$\Gamma_{post} \sim \Gamma_{prior} - \mathbf{C}_r\mathbf{D}_r\mathbf{C}_r^{T}. \tag{20}$$

where $\mathbf{C}_r$ represents the first $r$ columns of $\mathbf{C}$ and similarly for $\mathbf{D}_r$, with the rationale of neglecting posterior information in the directions where it has minimal effect.

## 2.5 Propagation of errors

Often of interest is how the observational data constrains outputs of a calibrated model, as opposed to how they constrain the calibrated parameters themselves. (A simple analogy is an extrapolation using a regression curve, which is generally of more interest than the regression parameters.) Such an output is termed a *Quantity of Interest* (QoI) $Q$, an example of which is the loss of ice volume above floatation (VAF), the volume of ice that can contribute to sea level, at a certain time horizon. Here we write $Q_T(\bar{c})$ to indicate the value of $Q$ based on the output of the calibrated model at time horizon $T$.

The distribution of $Q_T$ can be assessed by sampling from the posterior distribution of $\bar{c}$, although such sampling might be slow to converge. Alternatively an additional linear assumption can be made. Neglecting higher-order terms, $Q_T$ can be expanded around $\bar{c}_{MAP}$:

$$Q_T = Q_T(\bar{c}_{MAP}) + \left(\frac{\partial Q_T}{\partial \bar{c}}\right)(\bar{c} - \bar{c}_{MAP}). \tag{21}$$





As this is an affine transformation of a Gaussian random variable, $Q_T$ has a mean of $Q_T(\overline{c}_{MAP})$ and a variance of

$$\sigma^2(Q_T) = \left( \frac{\partial Q_T}{\partial \overline{c}} \right)^T \mathbf{\Gamma}_{post} \left( \frac{\partial Q_T}{\partial \overline{c}} \right) \tag{22}$$

If $\frac{\partial Q_T}{\partial \overline{c}}$ can be found at a number of times $T$ along a model trajectory, then the growth of uncertainty along this trajectory arising from parametric uncertainty can be assessed.

   Note the assumption of linearity in Eq. 21 is in general false due to the nonlinear momentum and mass balance equations

that define a time-dependent ice-sheet model. For the idealised experiments conducted in this paper, we compare the above estimate for the variance with that derived from sampling the posterior.

## 3   Numerical approach

In this study we use a new numerical code, `fenics_ice`. `fenics_ice` is a Python code which implements the time-dependent Shallow Shelf Approximation (SSA; MacAyeal (1989)). The SSA is an approximation to the complete Stokes stress

balance thought to govern ice flow. In the approximation the vertical stress balance is assumed to be hydrostatic, such that normal stress is in balance with the weight of the ice column. Additionally, flow is assumed to be depth-independent. These approximations reduce a three-dimensional saddle-point problem to a two-dimensional convex elliptic problem, enabling a more efficient solve. The nonlinear power-law rheology of the full Stokes problem is retained however. Despite these simplifications, the SSA describe flow of fast flowing ice streams and floating ice shelves well (Gagliardini et al., 2010; Cornford et al., 2020).

`fenics_ice` makes use of two sophisticated numerical libraries: FEniCS (Logg et al., 2012; Alnæs et al., 2015), an automated finite element method equation solver, and `tlm_adjoint` (Maddison et al., 2019), a library which implements automated differentiation of numerical partial differential equation solvers. FEniCS is a widely-used software library which abstracts the user away from low-level operations such as element-level operations. Rather, the weak form of the equation is written in Unified Form Language (UFL; Alnæs et al. (2014)), and FEniCS generates optimised low-level code which solves

the related finite-element problem with specified parameters (e.g. the order of the basis functions). `tlm_adjoint` is a library which implements high-level algorithmic differentiation of codes written with FEniCS or Firedrake.

   `tlm_adjoint` is used for several of the operations detailed in Section 2. It facilitates the minimization of the model-data misfit cost $J^c$ (Section ) with respect to $\overline{x}$ (which is equivalent to finding the mode of the posterior density of $\overline{x}$). The higher-order derivative capabilities of `tlm_adjoint` furthermore enable efficient computation of the product of the Hessian

of $J^c$ with arbitrary vectors, enabling an iterative eigendecomposition of the prior-preconditioned Hessian as described in 2.4. Finally, `tlm_adjoint`'s time-dependent capabilities enable differentiation of the temporal trajectory of the Quantity of Interest $Q_T$, enabling projections of posterior uncertainty as described in 2.5. In our experiments in the present study, our cost function $J^c$ is time-independent – but `tlm_adjoint` does allow for efficient calculation of Hessian-vector products for time-varying functionals (Maddison et al. (2019), their Section 4.2) – meaning time-varying data constraints can be considered

with `fenics_ice`. Currently `fenics_ice` calls SLEPc for the solution of the generalised eigenvalue problem

$$\mathbf{H}_{mis}\mathbf{C} = \mathbf{\Gamma}_{prior}\mathbf{C}\mathbf{\Lambda} \tag{23}$$





ensuring real-valued eigenvalues – though in future versions of `fenics_ice` randomized algorithms of the type used by Villa et al. (2018) can be used without loss of generality.

`fenics_ice` solves the Shallow Shelf Approximation by implementing the corresponding variational principle (Schoof, 2006; Dukowicz et al., 2010; Shapero et al., 2021):

$$
\int_\Omega 2H\nu\nabla\boldsymbol{\phi} : (\varepsilon_u + \mathrm{Tr}(\varepsilon_u)\mathbf{I})dA + \int_\Omega C^2\chi\boldsymbol{\phi}\cdot\boldsymbol{u}dA
$$
$$
+ \int_\Omega \mathcal{W}\nabla R\cdot\boldsymbol{\phi} - \mathcal{F}\nabla\cdot\boldsymbol{\phi}dA
$$
$$
+ \int_{\Gamma_c} (\boldsymbol{\phi}\cdot\boldsymbol{n})\cdot\left(\frac{1}{2}\rho g(H^2 - (\rho_w/\rho)|z_b^-|^2) - \mathcal{F}\right)dA = 0. \tag{24}
$$

Here $\boldsymbol{\phi}$ is a vector-valued test function, and $\boldsymbol{u}$ is the depth-integrated horizontal velocity vector. $\varepsilon_u$ is the horizontal strain-rate tensor $\frac{1}{2}(\nabla\boldsymbol{u} + \boldsymbol{u}^T)$, $\mathbf{I}$ is the 2×2 identity tensor, and ":" represents the Frobenius inner product. $H$ is ice-sheet thickness (the elevation difference between the surface, $z_s$, and the base, $z_b$). $\nu$ is ice viscosity, which depends on the strain rate tensor:

$$
\nu = \frac{1}{2}B\varepsilon_e^{\frac{1-n}{2n}},
$$
$$
\varepsilon_e = \varepsilon_{11}^2 + \varepsilon_{22}^2 + \varepsilon_{12}^2 + \varepsilon_{11}\varepsilon_{22}.
$$

$C$ is a spatially varying sliding coefficient, and $\chi$ is a function that indicates where ice is grounded according to the hydrostatic condition:

$$
H > (-\rho_w/\rho)R \equiv H_f \tag{25}
$$

where $\rho$ and $\rho_w$ are ice and ocean densities, respectively, and $R$ is bed elevation (note that $z_b = R$ when this condition is satisfied). In our code $C$ can in general depend locally on velocity and thickness, though in this study we consider only a linear sliding law, i.e. one in which $C$ varies only with location.

$\mathcal{F}$ is defined as

$$
\mathcal{F} = \begin{cases} \frac{1}{2}\rho gH^2 & H > H_f, \\ \frac{1}{2}\rho g(\delta H^2 + (1-\delta)H_f^2) & o.w., \end{cases} \tag{26}
$$

and $\mathcal{W}$ as

$$
\mathcal{W} = \begin{cases} \rho gH & H > H_f, \\ \rho gH_f & o.w. \end{cases} \tag{27}
$$

$\Gamma_c$ is defined as the calving boundary, i.e. the boundary along which the ice sheet terminates in the ocean (or in a cliff on dry land), and $\boldsymbol{n}$ is the outward normal vector at this boundary. Finally, $|z_b^-|$ indicates the negative part of the ice base, i.e. it is zero





when $z_b = R > 0$. The third integral of Eq. 24 is the weak form of the *driving stress* of the ice sheet, $\boldsymbol{\tau}_d = \rho g H \nabla z_s$. Although in our experiments in this study we consider only grounded ice, the full weak form is shown for completeness. The form of the driving stress term used here, $\nabla \mathcal{F} + \mathcal{W} \nabla R$, is not standard in glacial flow modelling, but it is equivalent to the more common form when thickness is represented by a continuous finite-element function.

In addition to the momentum balance, the continuity equation is solved:

$$H_t + \nabla \cdot (H\boldsymbol{u}) = b. \tag{28}$$

Here $b$ represents localised changes in mass at the surface or the base of the ice sheet, i.e. accumulation due to snowfall or basal melting of the ice shelf by the ocean (though in the present study, surface $b$=0). The continuity equation is solved using a first-order upwind scheme which is implicit in $H$ and explicit in $\boldsymbol{u}$.

We discretize velocity ($\boldsymbol{u}$) using 1st-order continuous Lagrange elements on a triangular mesh. In the present study thickness ($H$) is discretized with 1st-order continuous Lagrange elements as well – although we point out that formulation (Eq. 26 and Eq. 27), together with an appropriate discretization for the continuity equation (28), will allow for discontinuous Galerkin elements (which have been found in more realistic experiments with `fenics_ice` to improve stability of time-dependent simulations). Eq. 24 is solved for $\boldsymbol{u}$ with a Newton iteration, with the Jacobian calculated at the level of the weak equation

form using core `FEniCS` features. In the early iterations of the Newton solve, the dependence of $\nu$ on $\boldsymbol{u}$ is ignored in the Jacobian. This "linear" fixed-point iteration (often referred to in glacial modelling as Picard iteration, Hindmarsh and Payne (1996)) aids the Newton solver as it has a larger radius of convergence; once the Picard iteration is suitably converged, the full Newton iteration is applied.

To carry out an inversion, a cost function is minimized using the L-BFGS-B algorithm (Zhu et al., 1997; Morales and

Nocedal, 2011) supplied with SciPy 1.5.2 (although note that no bounds on the controls are used). SLEPc (Hernandez et al., 2005) is used to implement the eigendecomposition described in Section 2.4, using a Krylov-Schur method. Rather than solve the eigenvalue problem (Eq. 14), we solve the Generalised Hermitian Eigenvalue Problem

$$\mathbf{H}_{mis}\mathbf{C} = \Gamma_{prior}^{-1}\mathbf{C}\boldsymbol{\Lambda}, \tag{29}$$

which guarantees real-valued eigenvalues. (Despite $\Lambda$ being real-valued, the application of SLEPc to the non-hermitian eigen-

value problem Eq. 14 represents eigenvectors as imaginary, effectively doubling the memory requirements.)

## 4    Numerical Experiments

In this study, we aim to do the following:

1. Establish that control-method optimisations can be carried out with `fenics_ice`

2. Calculate eigendecompositions of the prior-preconditioned model-misfit Hessian as described in 2.4, examining the

impacts of regularisation, resolution, and spatial density and autocorrelation of observations $\breve{y}_{obs}$ on the reduction of variance in the posterior.





3. Propagate the posterior uncertainty on to a Quantity of Interest $Q_T$ as in 2.5

4. Establish, through simple Monte Carlo sampling, that the variance found through Eq. 22 is accurate.

Control method optimisations using ice-sheet models have been done extensively, with parameter sets of very high di-
mension (e.g., Cornford et al., 2015; Goldberg et al., 2015; Isaac et al., 2015), so our results regarding (1) above simply
demonstrate the capabilities of `fenics_ice` but are not novel. Isaac et al. (2015) carries out eigendecompositions of the
prior-preconditioned model-misfit Hessian and projects the associated uncertainty on to a Quantity of Interest – however, their
QoI is time-independent. Importantly, Hessian-based Uncertainty Quantification has not been implemented for a model of ice
dynamics using Algorithmic Differentiation before. Moreover, a time-dependent QoI has not been considered, nor has the
impact of observational data density on the posterior uncertainty.

To investigate these and similar factors comprehensively, as well as validate the assumption of Gaussian statistics that leads
to Eq 22, requires a model setup that is relatively inexpensive to run. We therefore choose one of the simplest frameworks
possible for our numerical experiments, that of the Benchmark experiments for higher-order ice sheet models (ISMIP-HOM)
intercomparison (Pattyn et al., 2008). We adopt the experiment ISMIP-C, a time-independent experiment in which an ice sheet
slides across a doubly-periodic domain with constant thickness and a basal frictional factor that varies sinusoidally in both
horizontal dimensions. The relation between velocity and basal shear stress is linear:

$$\boldsymbol{\tau}_b = -C^2(x, y)\boldsymbol{u} \tag{30}$$

where $C$ is the factor from the second integral of Eq. 24, which has the form

$$C^2(x, y) = 1000 + 1000 \sin\left(\frac{2\pi x}{L_x}\right) \sin\left(\frac{2\pi y}{L_y}\right) \tag{31}$$

with units of Pa (m/a)$^{-1}$, where $L_x$ and $L_y$ are experimental parameters. In this ISMIP-C specification, thickness is constant
($H$ = 1000 m) and a shallow surface slope of $0.1°$ is imposed. In this ISMIP-HOM intercomparison, SSA models compared
well with Stokes models for $L_x$, $L_y$ over $\sim$40 km, so this is the value we use in our study. A regular triangular mesh is used to
solve the model. Unless otherwise stated, cell diameter of the mesh is 1.33 km.

In our experiments, the momentum balance (Eq. 24) is solved on a highly refined grid and taken to be the "truth". To generate
synthetic observations, values are interpolated to predefined locations. Observational error is then simulated by adding Gaussian
random noise to these values. (These synthetic observations correspond to $\breve{u}_{obs}$ in Eq. 2.) In this study observational points
occur at regular intervals, though our code allows for arbitrary distributions of observation points. Unless stated otherwise, in
this study observational data points are spaced 2 km apart, with the velocity vector components coincident, and observational
uncertainties are mutually independent with a standard deviation of 1 m/a. The regular spacing of observational points is
not realistic and other studies use randomly scattered locations (e.g., Isaac et al., 2015); however, this choice is in line with
the idealised nature of our study and furthermore allows comprehensive investigation of the effects of observational density
(Section 5.4).

An inverse solution $\bar{c}$ is then found using a control method, where $\bar{c}$ is the vector of nodal coefficients of $C$. Below we refer
to $C$ as the *sliding parameter*. Our cost function $J_{IS}^c$ is composed of a misfit term equal to the negative log density of observed





velocities conditioned on $\overline{c}$ (*cf.* Eq. 2), and the regularisation operator is the discretised form of Eq. 9:

$$
\begin{aligned}
J_{IS}^c =& \frac{1}{2}\|\breve{\boldsymbol{u}}_{obs} - \breve{\boldsymbol{f}}(\overline{c})\|_{\boldsymbol{\Gamma}_{u,obs}^{-1}}^2 + \frac{1}{2}\int_\Omega \|\mathcal{L}(c - c_0)\|^2 dA \\
=& \frac{1}{2}\|\breve{\boldsymbol{u}}_{obs} - \breve{\boldsymbol{f}}(\overline{c})\|_{\boldsymbol{\Gamma}_{u,obs}^{-1}}^2 + \frac{1}{2}\|c - c_0\|_{\mathbf{L}\mathbf{M}^{-1}\mathbf{L}}^2
\end{aligned}
\tag{32}
$$

where $\mathbf{L}$ is as described in Section 2.3. In many studies, the optimal value for $\gamma$, the regularisation parameter, is determined heuristically through an $L$-curve analysis (e.g., Gillet-Chaulet et al., 2012). Here we examine, for different values of $\gamma$, the degree of uncertainty reduction associated with the cost-function optimisation. In other words, we seek the posterior density of $\overline{c}$, the coefficient vector of the finite-element function $C$. (We conduct an $L$-curve analysis, but only as a guideline for which values of $\gamma$ to examine.)

ISMIP-C does not prescribe a time-dependent component, but it is straightforward to evolve the thickness $H$ (which is initially uniform) according to Eq. 28, where $m = 0$. We define a Quantity of Interest $Q_T^{IS}$ as

$$
Q_T^{IS} = \int_\Omega (H(T) - H_0)^4 dA.
\tag{33}
$$

Unlike Volume above Floatation, the example given in Section 2.5, $Q_T^{IS}$ has no strong physical or societal significance. However, it is convenient to calculate and sufficiently nontrivial and nonlinear that the effects of uncertainty in $C$, as well as the strength of the prior covariance, can be seen. In our error propagation we evolve the ISMIP-C thickness for 30 years and use the time-dependent adjoint capabilities of `fenics_ice` to find $\partial Q_T^{IS}/\partial \overline{c}$ for discrete values of $T$ over this period, and uncertainty at these times is found using Eq. 22; an uncertainty "trajectory" is then found for $Q^{IS}$ via interpolation. Our results regarding the uncertainty of $Q^{IS}$, and the quadratic approximation inherent in Eq. 22, are then tested via sampling from the posterior as described in Section 5.3.1.

## 5 Results

### 5.1 Parameter uncertainties

#### 5.1.1 Effect of regularisation

An $L$-curve for our inversion results (Fig. 1) shows the behaviour of regularisation cost and model-data misfit as $\gamma$ is varied over 3 orders of magnitude. In all inversions, $\overline{c}$ is initialised assuming a point-wise balance between driving stress and basal drag arising from interpolated velocity observations, and $J^c$ is lowered from the initial value by a factor of $\sim 10^3$ (meaning the probability density associated with $C$, proportional to $e^{-J^c}$, is increased by a factor of approximately $10^{400}$).

While misfit does not vary greatly in a proportional sense, it suggests $\gamma = 10$ as a reasonable tradeoff between misfit and regularisation. Fig. 2 displays results of an inversion with a "strong" level of regularisation ($\gamma = 50$; referred to below as the $\gamma_{50}$ experiment). The resulting $C$ is relatively smooth (Fig. 2(a)), and the misfit is generally small though with some outliers (Fig. 2(d)). (Misfit is displayed as a histogram of errors – obtained by interpolating the finite-element solution to the sampled





velocity locations – rather than as a spatially continuous function, to emphasize the discrete nature of the model-data misfit.)

Fig. 3 gives equivalent results for a "weak" regularisation inversion ($\gamma = 1$; referred to below as the $\gamma_1$ experiment). The misfit distribution is similar but the inverted sliding parameter is significantly noisier, as a result of weaker constraints on these "noisy" modes by the prior.

The effect of regularisation on reduction of uncertainty can be seen from examining the eigenvalues defined by Eq. 14. More precisely, the ratio $1/(1 + \lambda_i)$, where $\lambda_i$ is the $i$th leading eigenvalue, is examined. As shown in Section 2.4, this ratio gives

the reduction in variance of the associated eigenvector in the posterior PDF relative to the prior distribution. In Fig. 4(a) this quantity is shown for the eigenvalue spectra corresponding to $\gamma$ = 1, 10, and 50. For all inversions, uncertainty reduction is several orders of magnitude for the leading eigenvalues, but the tails of the spectra are quite different. In the case of strong regularisation, there is little reduction in variance beyond $i \sim 100$, while in the weakly regularised case there is considerable reduction across the entire spectrum. This discrepancy can be interpreted as the prior providing so little information in the

low-regularisation case that the information provided by the inversion reduces uncertainty across all modes. The comparison of eigenvalue spectra across experiments is only meaningful to the extent that the corresponding eigenvectors are equivalent. A comparison between the four leading eigenvectors in the high- and low-regularisation experiments (Fig. 5) shows they are not equivalent but have the same overall structure. (Differences arise due to $\overline{c}_{MAP}$ but also due to differences in $\mathbf{\Gamma}_{prior}$.)

Approximating the posterior covariance of $\overline{c}$, $\Gamma_{post}$, also allows an estimation of $\Sigma_C$, the *pointwise* variance of $C$. This is

done via calculation of the square root of the diagonals entries of $\Gamma_{post}$, i.e. the standard deviation of the marginal distributions of the coefficients of $\overline{c}$. $\Sigma_C$ is shown for the inversions discussed above in Figs. 2(b) and 3(b). Pointwise uncertainties in $\gamma_1$ are 5-10 times larger than in $\gamma_{50}$. For $\gamma_{50}$ there is a clear pattern of higher uncertainty where the bed is weaker (i.e. $C$ is smaller), though for $\gamma_1$ it is difficult to discern any pattern.

### 5.1.2 Effect of resolution

The impacts of grid resolution on eigenvalue spectra are investigated (Fig. 4(b)). In Isaac et al. (2015), it was shown that the leading eigenvalues were independent of the numerical mesh, implying that the leading eigenvectors – the patterns for which uncertainties are quantified – are not dependent on the dimension of the parameter space (which would be an undesirable property). Our spectra suggest that at 2 km resolution, there is mesh dependence; but the spectra for 1.33 and 1 km resolution are in close agreement, suggesting mesh independence. Consistent values of $\gamma$ and $\delta$ are used for these experiments, meaning

the results of the $L$-curve in Fig. 1 are not dependent on model resolution.

### 5.2 Propagation of uncertainties

### 5.3 Linear propagation of uncertainties

The low-rank approximation of the posterior covariance of $\overline{c}$ found with Eq. 20 can be used to estimate the uncertainty of $Q_T^{IS}$ using the Eq. 22. To do so, $\frac{\partial Q_T^{IS}}{\partial \overline{c}}$ must be found, which is done using Algorithmic Differentiation of the time-dependent

model as described in Section 4. Figs. 2(e) and 3(e) show $\frac{\partial Q_{30}^{IS}}{\partial \overline{c}}$ arising from their respective inversions. There is small-scale





noise in the low-regularisation experiment ($\gamma_1$), but the general pattern and magnitude between the two gradients is similar, with strengthening of weak-bedded areas and weakening of strong-bedded areas both leading to an increase in the fourth-order moment of thickness. The gradient of $Q_T^{IS}$ with respect to $\overline{c}$ is found for intermediate values of $T$ over the 30-year interval, with $\sigma(Q_T^{IS})$ calculated at these times – which can then be linearly interpolated to find a trajectory of uncertainty. In our experiments

we find the gradient of $Q_T^{IS}$ every 6 years, but ... *say something about how it is not time-limited to find gradients on shorter intervals*. In Fig. 6(a) these trajectories are shown for the $\gamma_{50}$ and $\gamma_1$ experiments, plotted as a 1-$\sigma$ error interval around the calculated trajectory of $Q_T^{IS}$.

The trajectory of uncertainties for $\gamma_{50}$ and $\gamma_1$ can also be seen in Fig 6(b), compared against the trajectories of

$$\left( \left( \frac{\partial Q_T^{IS}}{\partial \overline{c}} \right)^T \mathbf{\Gamma}_{prior} \left( \frac{\partial Q_T^{IS}}{\partial \overline{c}} \right) \right)^{1/2} \tag{34}$$

i.e. the prior uncertainty linearly projected along the trajectory of $Q_T^{IS}$. This uncertainty measure is not physically meaningful as it depends on the calculated $\partial Q_T^{IS}/\partial \overline{c}$, which in turn depends on the inversion for $\overline{c}$ and the related trajectory of $Q_T^{IS}$ – and a random sample from the prior distribution of $\overline{c}$ is unlikely to yield such a trajectory. Still, it serves as a measure of decrease in uncertainty arising from the information encapsulated in the observations and model physics.

$Q_T^{IS}$ is greater in magnitude in the $\gamma = 1$ experiment than in the $\gamma = 50$ experiment at all times – and it can be seen from

the uncertainty of the $\gamma_1$ trajectory that this difference is statistically significant. The two experiments have differing (inverse) solutions, with the $\gamma_1$ inversion favoring a closer fit to noisy observations at the cost of small-scale variability in the inverse solution. Our quantity of interest (the fourth-order moment of thickness) is sensitive to this small-scale variability, so uniformity of trajectories of $Q_T^{IS}$ would not be expected. At the same time, the level of QoI uncertainty in the $\gamma_1$ trajectory relative to that of the $\gamma_{50}$ QoI uncertainty is much smaller than the relative magnitudes of the inversion uncertainties (*cf.* Figs 2(b), 3(b))

would suggest. This can be rationalised by considering Eq. 22: uncertainty in the QoI will depend on the extent to which uncertain parameter modes project on to the gradient of the QoI with respect to the parameters. While the $\gamma_1$ inversion results are overall more uncertain, the leading order modes are still constrained quite strongly. Thus, while $Q_T^{IS}$ is to a degree sensitive to small-scale variability it may still filter the most uncertain modes of the $\gamma_1$ inversion, resulting in a smaller QoI uncertainty than expected. In fact, it can be seen from Fig. 6(b) that despite the large differences in prior distributions between $\gamma_{50}$ and $\gamma_1$,

the projections of the respective prior covariances along the trajectory of $Q_T^{IS}$ are very similar, suggesting that the gradient of $Q_T^{IS}$ does not project strongly on the modes which are poorly constrained in the $\gamma_1$ experiment.

The sensitivity of QoIs to small-scale variability is significant because not all glaciologically motivated QoIs are expected to have such sensitivities. For instance, the QoI considered by Isaac et al. (2015) was a contour integral of volume flux over the boundary of the domain, equivalent to a rate of change of ice volume – and such a quantity might be less sensitive to velocity

gradients and small-scale thickness change in the domain interior. On the other hand, a forecast focused on the impact of evolving surface elevation on proliferation of surface lakes, or on surface fractures, might be very sensitive to such variability. Therefore, when considering parametric uncertainty, it should also be considered whether the nature of this uncertainty impacts the uncertainty of the intended Quantity of Interest. *maybe move to discussion*





### 5.3.1 Direct sampling of QoI uncertainties

Ideally, the assumptions implicit in the calculation of QoI uncertainties shown in Fig. 6(a) would be tested through unbiased sampling from the prior distributions of $\overline{c}$; followed by using the sampled parameters to initialise the time-dependent model and generating a sample of trajectories of $Q_T^{IS}$; and finally scaling the probability of each member of the ensemble based on the observational likelihood function $p(\check{\boldsymbol{u}}_{obs}|\overline{c})$. However, given the dimension of the space containing $\overline{c}$ (equal to 900 in our idealised experiment; but on the order of $10^4$-$10^5$ in more realistic experiments), the number of samples required to ensure

nonnegligible likelihoods would not be tractable without a sophisticated sampling strategy such as Markov Chain Monte Carlo (MCMC) methods (Tierney, 1994) (and even then may require approximations similar to those described above (Martin et al., 2012)). However, such approaches are beyond the scope of this study.

The assumptions in our propagation of observational and prior uncertainty to Quantity of Interest uncertainty are ($i$) that of Gaussianity of the distribution of $\overline{c}$ and ($ii$) that of linearity of the map from $\overline{c}$ to QoI. While ($i$) cannot be tested for the reasons

stated above, ($ii$) can be tested by sampling from the calculated posterior distribution of $\overline{c}$, initialising the time-dependent model, and finding the ensemble variance and standard deviation of $Q_T^{IS}$. Our strategy for sampling from the posterior is described below, and is based on the derivation in Bui-Thanh et al. (2013).

A randomly sampled vector $\overline{x}$ will have covariance $\boldsymbol{\Gamma}_{post}$ and mean $\overline{c}_{MAP}$ if it is generated via

$$\overline{x} = \overline{c}_{MAP} + \mathbf{K}\overline{N} \tag{35}$$

where $\overline{N}$ is a sample from a multivariate normal distribution $\mathbb{N} \sim \mathcal{N}(0,\mathbf{I})$ of the same dimension as $\overline{c}$, and $\mathbf{K}$ is such that $\mathbf{K}\mathbf{K}^T = \boldsymbol{\Gamma}_{post}$. Hence it is required to find a suitable $\mathbf{K}$. We restate the generalised eigenvalue problem $\mathbf{H}_{mis}\mathbf{C} = \boldsymbol{\Gamma}_{prior}^{-1}\mathbf{C}\boldsymbol{\Lambda}$. Since $\mathbf{C}$ is orthogonal in the inverse prior covariance (*cf.* Eq. 16), the identity matrix $\mathbf{I}$ can be spectrally decomposed in $\overline{c}_i$ (the columns of $\mathbf{C}$):

$$\left( \sum \overline{c}_i \overline{c}_i^T \boldsymbol{\Gamma}_{prior}^{-1} \right) = \mathbf{I}. \tag{36}$$

Rearranging gives $\sum \overline{c}_i \overline{c}_i^T = \boldsymbol{\Gamma}_{prior}$, and so (*cf.* Eq. 20)

$$\begin{aligned}
\boldsymbol{\Gamma}_{post} &\sim \boldsymbol{\Gamma}_{prior} - \mathbf{C}_r \mathbf{D}_r \mathbf{C}_r^T \\
&= \sum_{i=1}^{n} \overline{c}_i \overline{c}_i^T - \sum_{i=1}^{r} \overline{c}_i \overline{c}_i^T \left( \frac{\lambda_i}{1+\lambda_i} \right) \\
&= \sum_{i=r+1}^{n} \overline{c}_i \overline{c}_i^T + \sum_{i=1}^{r} \overline{c}_i \overline{c}_i^T \left( \frac{1}{1+\lambda_i} \right).
\end{aligned} \tag{37}$$

We define the matrix $\mathbf{B}$:

$$\begin{aligned}
\mathbf{B} &= \boldsymbol{\Gamma}_{prior} + \sum_{i=1}^{r} \overline{c}_i \overline{c}_i^T \left( \frac{\lambda_i}{\sqrt{1+\lambda_i}} - 1 \right) \\
&= \sum_{i=r+1}^{n} \overline{c}_i \overline{c}_i^T + \sum_{i=1}^{r} \overline{c}_i \overline{c}_i^T \left( \frac{1}{\sqrt{1+\lambda_i}} \right).
\end{aligned} \tag{38}$$





And due to the $\mathbf{\Gamma}_{prior}^{-1}$-orthogonality of $\mathbf{C}$,

$$\mathbf{B}\mathbf{\Gamma}_{prior}^{-1}\mathbf{B}^T = \sum_{i=r+1}^{n} \bar{c}_i \bar{c}_i^T + \sum_{i=1}^{r} \bar{c}_i \bar{c}_i^T \left( \frac{1}{1+\lambda_i} \right)$$

$$= \mathbf{\Gamma}_{post}. \tag{39}$$

Therefore a suitable $\mathbf{K}$ is given by (*cf.* Eq. 11)

$$\mathbf{B}\mathbf{\Gamma}_{prior}^{-1/2} = \mathbf{B}\mathbf{L}\mathbf{M}^{-1/2}. \tag{40}$$

The action of the square root of the mass matrix $\mathbf{M}$ is found by a Taylor series approach (Higham (2008), eq. 6.38). Fig. 7 shows a result of sampling from the posterior in the $\gamma_{50}$ experiment. To the left (panel (a)), a realisation of the prior distribution, with mean zero and covariance $\mathbf{\Gamma}_{prior}$ is displayed. (This realisation is found similarly to that of the posterior, with the formula

$\overline{N}\mathbf{\Gamma}_{prior}^{1/2}$.) To the right (panel (b)), a realisation of the posterior distribution is shown with the mean $\bar{c}_{MAP}$ removed. (Note that both samples are derived from the same realisation of $\mathbb{N}$.) From comparing the images it can be seen that variance is greatly reduced, particularly at medium-to-large scales. By contrast, when the posterior distribution of $\gamma_1$ is sampled, the result is very similar to the prior. Very little reduction of variance is visually apparent, especially at small scales.

Using this method of sampling the posterior, an ensemble of 1,000 30-year runs is carried out for both low and high regu-
larisation experiments ($\gamma_1$ and $\gamma_{50}$, respectively), and standard deviations of $Q_T^{LS}$ are calculated at discrete times. Values are plotted in Fig. 6(b). (For each such calculation, the variance quickly converged to the value shown, so it is unlikely that the Quantity of Interest is under-sampled.) In the $\gamma_{50}$ experiment there is strong agreement between the sampled uncertainties and those found via projecting $C$ uncertainty along the linearised QoI trajectory, suggesting the linear approximation inherent in Eq. 22 is appropriate. In contrast, there are large discrepancies in the $\gamma_1$ case. It is likely that the small-scale noise inherent
in the low-regularisation samples (*cf* Fig. 8) impacts the Quantity of Interest strongly enough that the linear approximation in Eq. 22 breaks down – despite that this noise does not strongly affect the cost function $J^c$. As mentioned in Sec. 5.2, this may be due to the nature of the QoI. In the $\gamma_{10}$ experiment (not shown), the disagreement in uncertainties is on the order of 30 – greater than for the $\gamma_{50}$ experiment but far less than for $\gamma_1$.

## 5.4 Observational density and uncertainty

In all results presented to this point, the imposed locations observational data $\breve{u}_{obs}$, $\breve{v}_{obs}$ lie on a regular grid with a spacing of 2 km. Here we consider the effects of the observational spatial density on the reduction of uncertainty in $\bar{c}$.

### 5.4.1 Effect of observation spacing

Eigendecompositions of the prior-preconditioned misfit Hessian ($\hat{\mathbf{H}}_{mis}$) are carried out for observational spacings of 500 m, 1 km, 2 km, 4 km, and 8 km. (The 2 km case corresponds to the $\gamma_{10}$ experiment in Fig. 4(a).) In all other respects the experiments
are identical. Results are shown for comparison in Fig. 4(c). Increasing spatial density appears to reduce uncertainty: in the 500 m case, there is considerable uncertainty reduction even in cases where there is almost no reduction in coarser-observation





cases. The result is intuitive: each increase in observational density quadruples the number of independent constraints, effectively adding more information (though a more sophisticated framework is required to quantify the information increase from a given observation, *e.g.*, Alexanderian et al. (2014)).

Comparison of eigenspectra relies on the corresponding eigenvectors being the same, or similar, between the experiments. As in the regularisation and resolution experiments, the eigenvectors depend on the exact form of $\hat{\mathbf{H}}_{mis}$ which in turn depends on $\overline{c}_{MAP}$, which may differ between the experiments due to the differing number of points. However they are likely to be of similar patterns (on the basis of the results of Section 5.1.1).

### 5.4.2   Effect of observational covariance

The results described above imply that posterior uncertainty could be made arbitrarily small by increasing the spatial density of observations (although we do not examine observations more dense than 500 m). However, the decreasing uncertainty relies on the observations being statistically *independent*, which is unlikely to be the case as observations become more and more dense. We consider here the implications of a nonzero spatial covariance. Rather than imposing a realistic observational covariance matrix, we consider a simplified covariance structure in which correlations decay isotropically. That is, our observational
covariance matrix $\mathbf{\Gamma}_{u,obs}$ is given by

$$\mathbf{\Gamma}_{u,obs}(i,j) = \begin{cases} \sigma^2_{u,obs} & i = j, \\ \sigma^2_{u,obs}\, e^{-\frac{|\boldsymbol{x}_i - \boldsymbol{x}_j|^2}{d^2_{auto}}} & o.w. \end{cases} \tag{41}$$

    Here $\sigma_{u,obs}$ is the observational uncertainty and $\boldsymbol{x}_i$ is the position of observation $i$. (By contrast, $\mathbf{\Gamma}_{u,obs}$ in all experiments described above is a diagonal matrix with entries $\sigma^2_{u,obs}$.) A value of 1 m/a is used for $\sigma_{u,obs}$, as in all previous experiments; and $d_{auto}$ is set to 750 m. We assert that the observations of orthogonal velocity components are independent, i.e. $\mathbf{\Gamma}_{u,obs}$ is
block-diagonal with each block corresponding to a velocity component. While velocity component uncertainties are likely to correlate, introducing spatial correlation among the individual components already greatly changes the effect of observation spacing on uncertainty reduction, as seen in Fig. 4(d). When observational spacing is large compared to $d_{auto}$, an increase in density has a similar effect to that seen in the zero-spatial correlation case (Fig. 4(c)). But for observational spacing on the order of $d_{auto}$, additional observations have minimal effect.

## 6   Discussion and Conclusions


The inversion of surface velocities for basal conditions is ubiquitous in ice-sheet modelling – but in most studies in which this is done, the uncertainty of the resulting parameter fields is not considered, and the implications of this parametric uncertainty on projection uncertainty is not quantified. We introduce `fenics_ice`, a numerical Python code which solves the Shallow Shelf Approximation (SSA) for ice-sheet dynamics. The code uses the `FEniCS` library to facilitate finite-element solution of
partial differential equations. Algorithmic differentiation is implemented with the `tlm_adjoint` library, allowing for adjoint





generation of the time-dependent and time-independent versions of the SSA. This feature is used to aid in inversions of surface velocity for parameter fields such as the basal sliding parameter. In addition, the `tlm_adjoint` library allows efficient second-order differentiation of the inversion cost function, allowing a low-rank approximation to the cost function Hessian. We utilise this ability to exploit the connection between the control-method inversions typically carried out with ice-sheet models, and a Bayesian characterisation of the uncertainty of the inverted parameter field. This interpretation allows us to form a local approximation to the posterior probability density at the maximum a posteriori (MAP) point. With a time-dependent Quantity of Interest (QoI) which depends on the outcome of the inversion, the adjoint features of `fenics_ice` allow linear propagation of parametric uncertainty to QoI uncertainty.

We apply our framework to a simple idealised test case, Experiment C of the ISMIP-HOM intercomparison protocol, involving an ice stream sliding across a doubly periodic domain with a varying basal friction parameter. An idealised time-varying QoI is defined, equivalent to the fourth moment of thickness in the domain, as thickness evolves due to mass continuity. The posterior probability density is examined, suggesting mesh independence (provided resolution is high enough). It is shown that the level of uncertainty reduction relative to the prior distribution depends on the amount of information in the prior (or, equivalently, the degree of regularisation). Uncertainty of the QoI is found along its trajectory, and is found to increase with time and also found to be larger with less-constrained priors. However, the difference in the uncertainty of the QoI is far less than that of the parametric uncertainty, due to filtering of high-frequency modes. Testing the validity of our local approximation of the posterior probability density is beyond the scope of our study. However, sampling from our posterior allows us to test the linearity of the parameter-to-QoI mapping, and this approximation is seen to be accurate with a moderately strong prior.

A key difference between our approach and the control-method inversions typically undertaken is the Euclidean inner product that appears in the misfit component of the cost function, as opposed to an area integral of velocity misfit. As discussed in Section 2.3, the latter formulation leads to difficulties with a Bayesian interpretation by conflating the observational error covariance with mesh-dependent factors. In our study observation locations are imposed on a regular grid. It is shown that, with statistically independent observations, posterior uncertainty is continually reduced as the observational grid becomes more dense. When a spatial correlation of observations is considered, however, there is little reduction of uncertainty when adding observations beyond a certain spatial density. This result is of significance to ice-sheet modelling: most ice-sheet model studies which calibrate parameters to velocity observations (including those mentioned in the Introduction) do not consider the spatial correlation of observations. As discussed in Section 2.3, these studies express the model-data misfit as an area integral – meaning that, effectively, observations in adjacent model grid cells are considered independent. If grid cells are sufficiently large, this is likely a suitable approximation – though with higher and higher resolutions being used in ice-sheet modelling studies (Cornford et al., 2013), it should be considered whether the spatial covariance of observations is such that it might affect results. Assessing such effects poses an additional challenge, however, as ice-sheet velocity products are not generally released with spatial error covariance information (Rignot et al., 2017).

Our study does not consider "joint" inversions, i.e. inversions with two or more parameter fields. With such inversions, complications can arise when both parameters affect the same observable, potentially leading to equifinality/ill-posedness. An example of such a pair is $C$, the sliding coefficient, and $B$, the ice stiffness in the nonlinear Glen's rheology (*cf.* Eq. 24),





which can both strongly affect ice speeds in a range of settings. The version of `fenics_ice` presented in this study is not capable of joint inversions or of Hessian-vector products with multiple parameter fields. However, the technical hurdles are minor. More importantly, though, Hessian-based Bayesian uncertainty quantification with multiple parameter fields has not, to our knowledge, been carried out in an ice-sheet modelling context, and may present difficulties due to a larger problem space

or the equifinality issues mentioned above. (Instead of performing a joint inversion, Babaniyi et al. (2020) use a Bayesian Approximation Error framework, treating the stiffness parameter as a random variable.) Nonetheless, the investigation of joint inversions and uncertainty quantification is a future research aim for `fenics_ice`.

Model uncertainty is not accounted for in our characterisation of parametric uncertainty. In the expression for the posterior probability density (Eq. 4), the model-misfit term is expressed as the difference between observed and modelled velocity, and

the uncertainty is assumed to arise from the observation platform. In fact, the discrepancy between modelled and observed velocity is the sum of observation error, $\epsilon_{obs}$ and model error, $\epsilon_{model}$. This second error source can be considered a random variable, as it arises from incomplete knowledge about the ice-sheet basal environment and material properties of the ice, as well as the approximations inherent in the Shallow Shelf Approximation. Characterising this uncertainty is challenging as it requires both perfect knowledge of the basal sliding parameter and observations with negligible error, and is beyond the scope

of our study. Future research, however, could involve using a model which implements the full Stokes solution (e.g., Gagliardini et al., 2010) to partially characterise this uncertainty.

Our study does not consider time-dependent inversions, i.e. control methods where the cost function is time-dependent. While the majority of cost-function inversions are time-independent, there are a growing number of studies carried out with time-dependent inversions (Larour et al., 2014; Goldberg et al., 2015) and it is possible that such methods may provide lower

uncertainty in calibration of hidden parameters (simply by providing additional constraints) and hence in ice-sheet projections. `fenics_ice` (or rather `tlm_adjoint`) is capable of eigendecomposition of Hessian matrices of time-dependent cost functions (Maddison et al., 2019), but time-dependent Hessian-vector products are computationally expensive, requiring checkpointing and recomputation of both forward and reverse mode model information, and it is unlikely that full eigenvalue spectra can be found for even modestly sized problems. It is hopeful that for realistic problems of interest only a small fraction

of eigenvalues will need to be found to accurately approximate the posterior covariance, but more work is required in this area.

*Code availability.*    The `fenics_ice` code can be obtained from https://doi.org/10.5281/zenodo.4633106 and is freely available under the LGPL-3.0 License. The branch containing the version of the code used for this manuscript is *GMD_branch*. Python scripts for running all experiments and creating all figures in this manuscript can be found in the *example_cases* directory, and installation instructions for `fenics_ice` and dependencies can be found in the *user_guide* folder. The commit tag of `tlm_adjoint` used for the experiments in this

manuscript is 79c54c00a3b4b69e19db633896f2b873dd82de4b.



*Author contributions.* CK and JT developed the `fenics_ice` software with considerable support from JM. DG and JM developed the mathematical framework implemented in `fenics_ice`. DG wrote the manuscript and ran and analysed the experiments.

*Competing interests.* The authors declare they have no competing interests.

*Acknowledgements.* The authors acknowledge NERC Standard Grants NE/M003590/1 and NE/T001607/1 (QUoRUM).





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



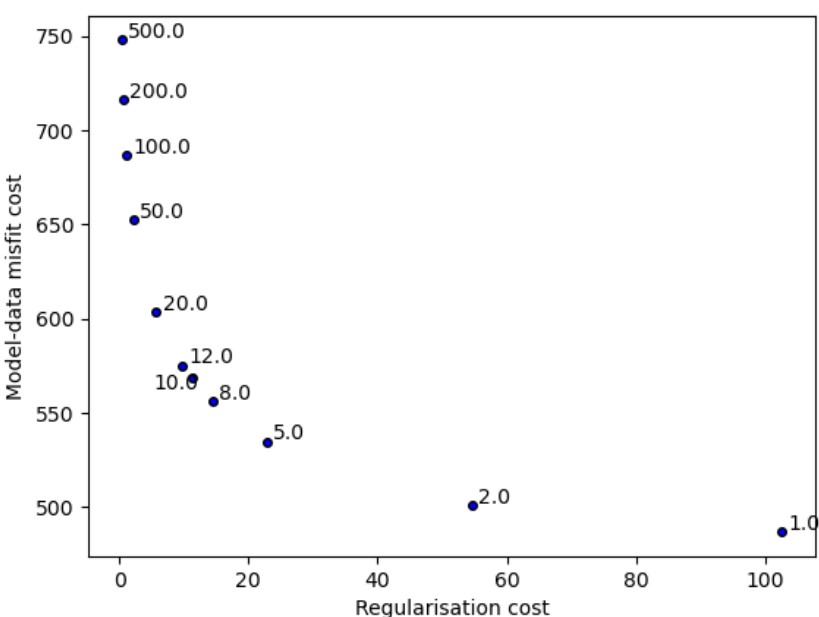

**Figure 1.** An $L$-curve showing the tradeoff between model misfit (first and second terms of Eq. 32) and the regularisation cost (the second terms of Eq. 32 divided by $\gamma$). Associated values of the regularisation parameter $\gamma$ are shown. In all optimizations, $\delta$ is equal to $10^{-5}$ and observational points occur at intervals of 2 km.



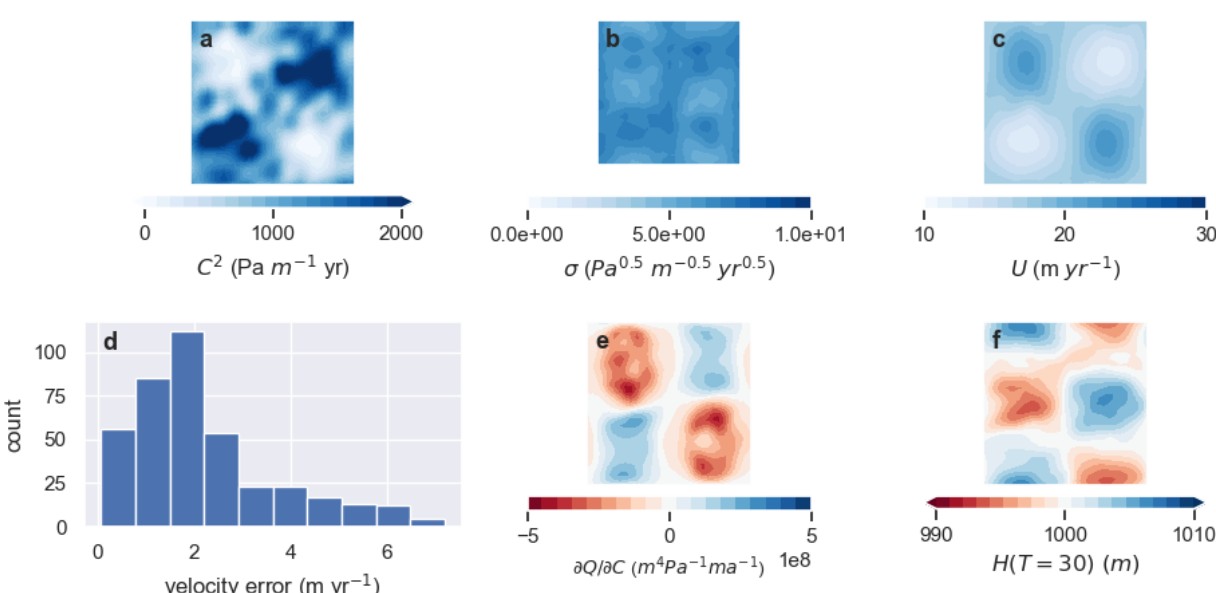

**Figure 2.** Results of the control-method inversion with $\gamma = 50$. (a) The recovered basal traction $C^2$. (b) The point-wise standard deviation of the sliding parameter $C$. (c) The surface speed associated with the inverted $C$. (d) Histogram of model-data velocity misfit (where misfit is the 2-norm of the difference of observed and modelled velocity). (e) The sensitivity of $Q_{30}^{IS}$ to the sliding parameter. (f) Thickness after 30 years of time stepping.

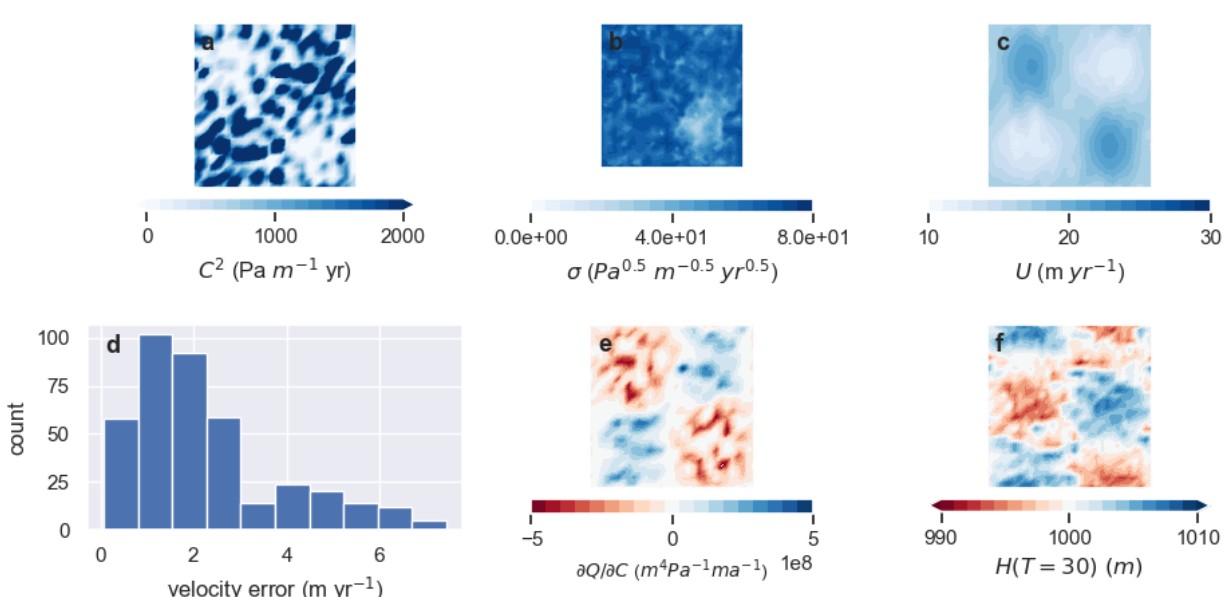

**Figure 3.** Similar to Fig. 2 but with $\gamma = 1$. Note the difference in colormap with Fig. Fig. 2(b).





**Figure 4.** Uncertainty reduction factor $1/(1+\lambda_k)$ versus eigenvalue index $k$ for a range of experimnents. (a) Dependence of reduction spectra on regularisation parameter $\gamma_\alpha$. (b) Dependence of reduction spectra on model resolution. (c) Dependence of reduction spectra on density of observational sample points. (d) Dependence of reduction spectra on density of observational sample points with nonzero observational covariance.

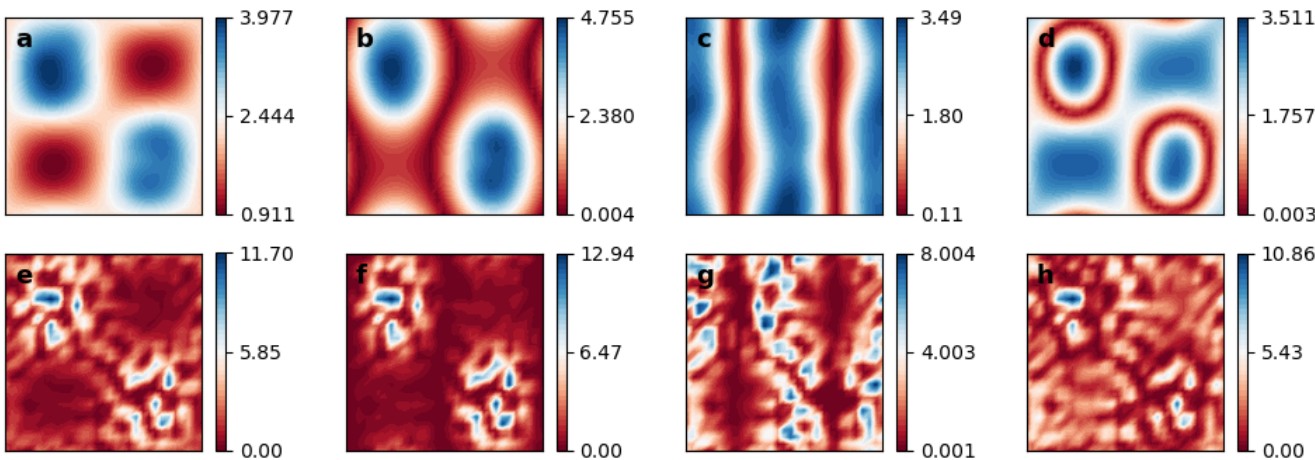

**Figure 5.** (a)-(d): Leading four eigenvectors of $C$ in $\gamma_{50}$ experiment. (e)-(g): Leading four eigenvectors of $C$ in $\gamma_1$ experiment.

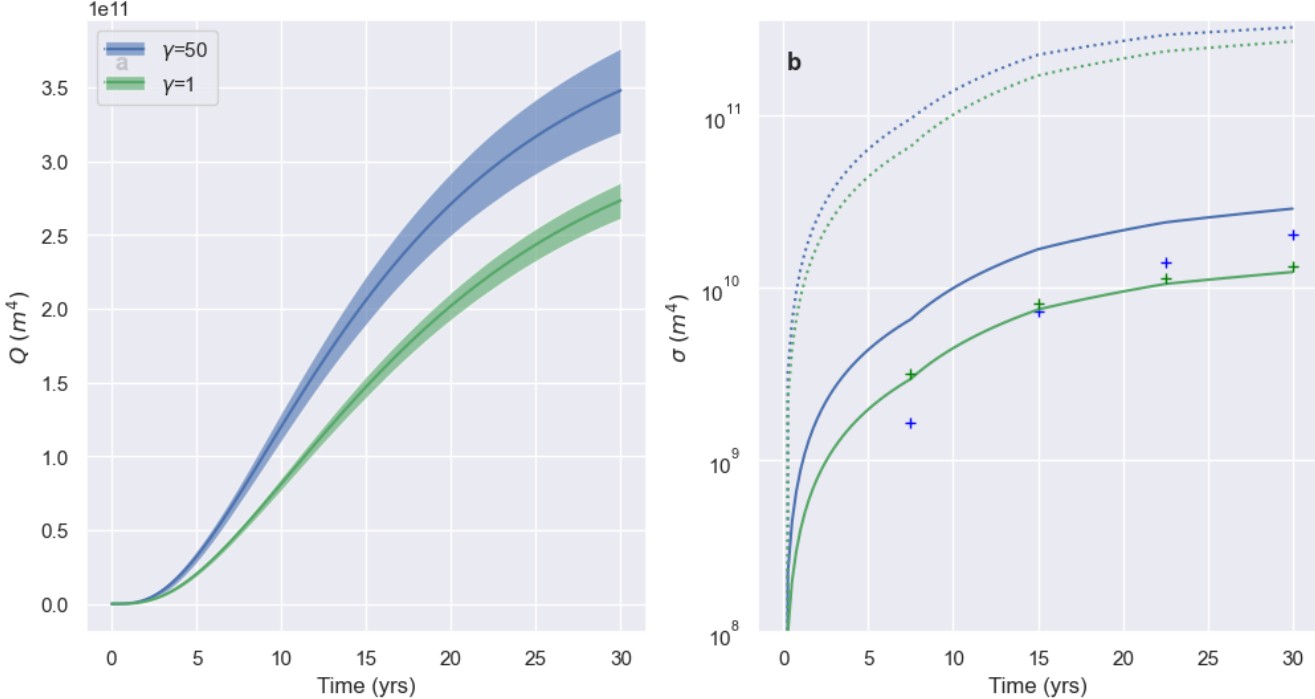

**Figure 6.** (a) Paths of $Q_T^{IS}$ in $\gamma_{50}$ experiment (blue) and $\gamma_1$ experiment (green). Shading shows the 1-$\sigma$ uncertainty interval for each trajectory calculated by projecting the Hessian-based (posterior) uncertainty along the linearised trajectory. (b) Uncertainties in the time-dependent experiments over time. Dashed lines: prior uncertainties projected along the linearised $Q_T^{IS}$ trajectory. Solid lines: Hessian-based posterior uncertainties projected along the linearised trajectory. Markers: Standard deviation of $Q_T^{IS}$ from sampling the posterior density. In both panels, green corresponds to $\gamma_{50}$, and blue to $\gamma_1$.



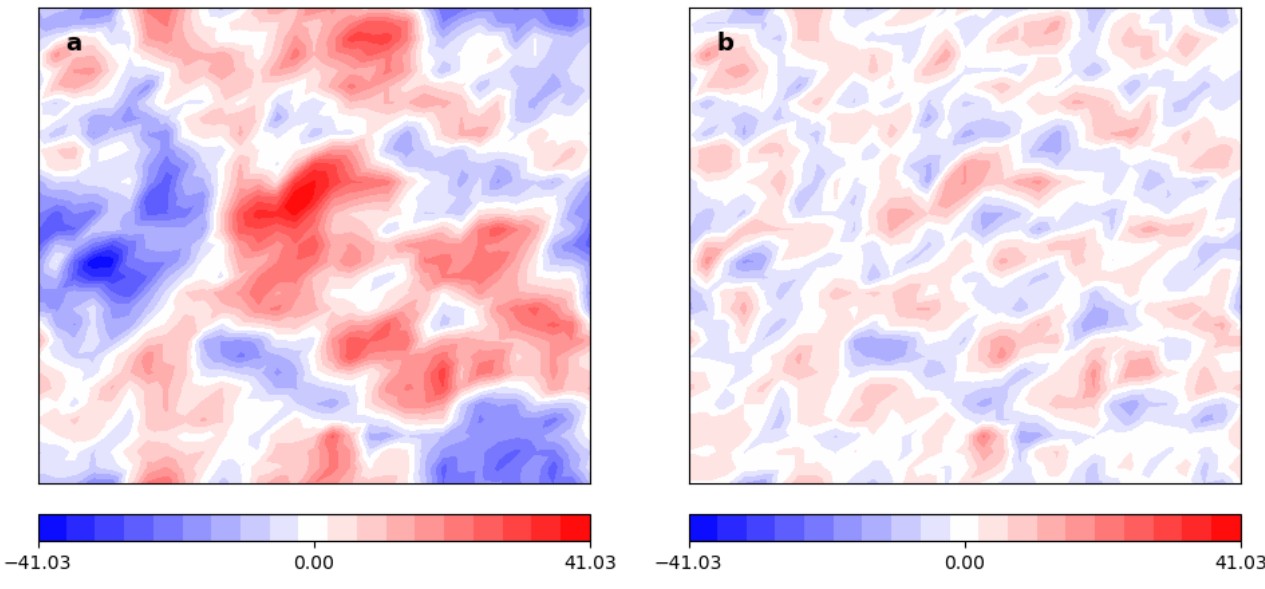

**Figure 7.** (a) A realisation of the prior density of $C$ for the $\gamma_{50}$ experiment. (b) A realisation of the posterior density of $C$ for the $\gamma_{50}$ experiment with mean $\overline{c}_{MAP}$ removed.



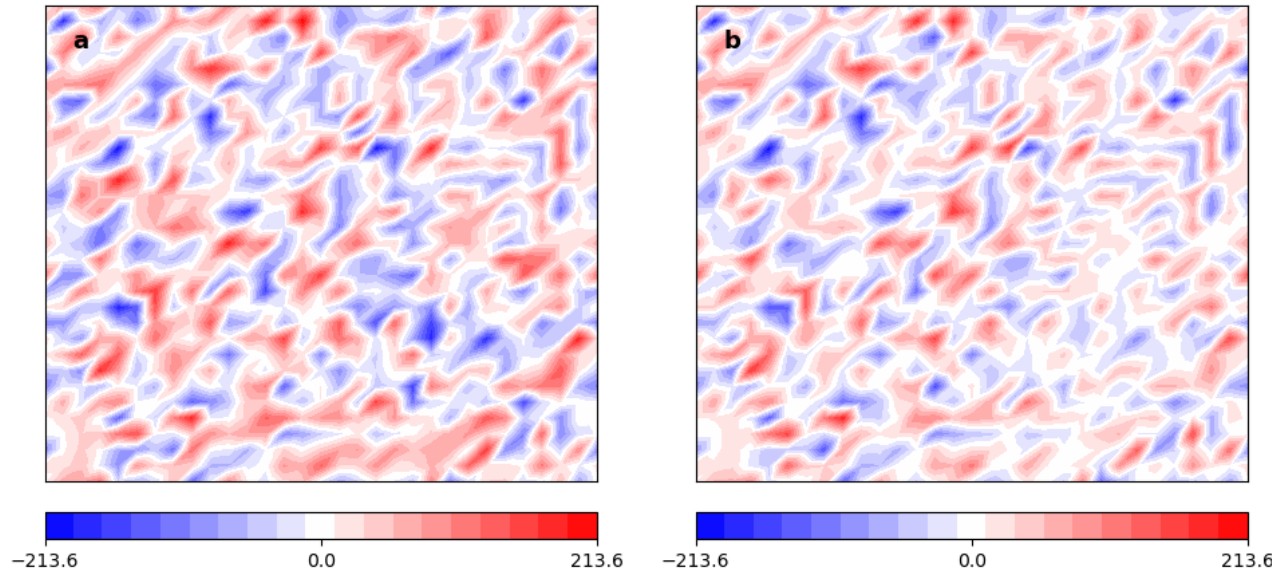

**Figure 8.** (a) A realisation of the prior density of $C$ for the $\gamma_1$ experiment. (b) A realisation of the posterior density of $C$ for the $\gamma_1$ experiment with mean $\bar{c}_{MAP}$ removed. Note the difference in colormap with Fig. 7.