# Peer review of "fenics\_ice 1.0: A framework for quantifying initialisation uncertainty for time-dependent ice-sheet models"

_Geoscientific Model Development, 2021_

## Author Comment (AC2)

The authors would like to express their thanks to both referees for providing such thoughtful comments on our manuscript -- and also to the Associate Editor for overseeing our manuscript.

We apologise for the length of time required for our response and resubmission. The delay was primarily as a result of our implementation of the Gauss-Newton Hessian approximation as asked after by Rev. 2, which was nontrivial but we feel makes a contribution to the manuscript and to the code.

Line numbers referenced below refer to the revised version (not the difference report or the original version)

**Daniel N Goldberg on behalf of authors**

**Comments from Reviewer 1**

The uncertainty in model data is propagated to a Quantity of Interest (QoI) in ice sheet simulations using a Bayesian approach. This kind of analysis has been published previously but the authors extend it here to time dependent QoIs. They show how to reduce the uncertainty in the prior by utilizing Bayes' formula for the posterior. The method is tested with the SSA ice sheet equations solving an ISMIP-HOM problem. Parameters are varied in numerical computations. Examples of possible future work are given in the final section. The code for the experiments is available for free.

**We thank the reviewer for this synopsis. We also point out that (to our knowledge) the application of Algorithmic Dlfferentiation to this problem (UQ of ice-sheet initialisation) is new.**

The paper is suitable for GMD and is improved if the comments below are addressed somehow in a revised version.

**General comments**

Sect. 2.4: \$\Gamma\_{post}\$ depends on how \$r\$ is chosen. If there is a gap in the eigenvalue distribution then \$r\$ can be chosen such that the gap is between \$\lambda\_r\$ and \$\lambda\_{r+1}\$. But in general there is no gap. How should \$r\$ be chosen in a general case? This question is related to the choice of \$\gamma\$, see Fig 4a. If we believe in prior data then \$\gamma\$ should be large but if we don't then give data lower weight (or should \$\gamma\$ be viewed only as a regularization parameter?).

We agree the "choosing" or \$r\$ was a point not addressed. In this work the problems are small enough for us to recover the entire spectrum but in general this needs to be decided. Isaac et al (2015) suggest a strategy of ensuring lambda\_r is O(1). However a more pragmatic strategy we are considering is to choose \$r\$ so that eq (22) (the

variance of the Qol) has negligible change as \$r\$ increases. We mention both strategies now.

Choosing an appropriate value for the prior parameter \$\gamma\$ is in general very important beyond the scope of this idealised study. We explain (line 347) that, in line with other non-probabilistic inverse modelling studies, we carry out an L-curve analysis, but only as a guideline for choosing values of \gamma to examine. In general, it does have meaning, and we feel the background sections of the manuscript express this sufficiently.

Sect. 4, (33): Why is this particular Qol chosen? A bit longer motivation would be welcome.

We now give more motivation for this choice -- and also "call forward" to a passage that is (now) in the discussion (line 562) about the choice of Qol and its sensitivity to small scale variability.

Sect. 4: How is the prior \$c\_0\$ chosen? Maybe this is mentioned somewhere but it could be repeated here. A discussion of how to select \$c\_0\$ and its impact on the posterior result would be interesting.

**We now say:**

"In our experiments \$C\_0\$, the prior value of \$C\$, is uniformly zero -- indicating we assume no preconceieved notion of its mean value, only its spatial variability (implied by \$\gamma\$)."

It is beyond the scope of this idealised experiment, but in general C\_0 should depend on independent assessments e.g. the bed elevation \$R\$ may be sought as an unknown, but independent assessments exist. However, we do not mention this as we do not introduce additional unknown fields in our simple idealised framework.

Sect. 5.3: One would expect that the linear approximation of QoI should work for sufficiently small perturbations. Maybe the perturbations are too large when the regularization is small (\$\gamma\_1\$) and for smaller perturbations the approximation will work.

Yes, we agree -- and actually we feel this is echoed in our passage: "It is likely that the small-scale noise inherent in the low-regularisation samples (\textit{cf} Fig. 8) impacts the Quantity of Interest strongly enough that the linear approximation in Eq. 22 breaks down "

Sect. 6: The issues above with choice of \$r, \gamma, c\_0\$ and QoI could also be discussed in the last section.

**Thank you for the suggestion. We have decided to address these issues in the relevant sections of the methodology (see passages and line number references above)**

Specific comments

line 93: Euclidean inner product of a with  $\operatorname{A}_{-1}_{obs}^{9}$ . Next line  $|a|_{Gamma^{-1}_{obs}}^{9}$ .

**Thank you, fixed!**

105: Is there a weight missing in front of the prior term? Maybe \$\gamma\$?

This is as we intended it. Parameters such as \$\gamma\$ are meant to be included in \Gamma\_{prior} ie. see eq 11.

110: Tell that you maximize over  $\bar{c}$ .

**Done (though actually we minimize the negative log posterior)**

145: Define  $\frac{L}$  is defined for a function in (9). In (10)  $\frac{L}$  is applied to a vector c, at least c is a vector after the last equality in (10). Should  $J_{reg}^c$  depend on a weight too?

Thank you for pointing out this is undefined, we now make clear that \$\gamma\$ and \$\delta\$ are scalars (and thus \$\delta(\cdot)\$ implies multiplication). You also raise an important point, \$\gamma\$ and \$\delta\$ in general can vary spatially, though in the present they are treated as constant. We now write:

"where \$\gamma\$ and \$\delta\$ are positive scalars which are in general spatially varying, though in the present study we consider only constants."

These "weights" however are absorbed into the definition of \$\mathbf{L}\$.

149: Should the definition of \$L\$ have only one  $\lambda L$  in the integral? What are the bars over  $\phi$ ?

You are correct, apology for the oversight which is fixed. As described in 2.1, the overbar is our convention for a coefficient vector of a finite element function.

171: Mention that the eigenvalues are ordered such that  $\lambda_i^i = \frac{i+1}{$ .

**Done.**

179: leading eigenmodes -> leading eigenmodes with large eigenvalues.

With our implementation of the previous suggestion (that the eigenvalue ordering is in descending order), this is now implied, so we do not make this change.

242: Specify parameters \$B\$ and \$n\$

We are not sure if you meant to define these or give the values we used, but we now do both (lines 255-257)

251, 253: Specify \$\delta\$ which is different from \$\delta\$ in (9). Tell what o.w is.

Apologies, \delta is a symbol commonly used for this quantity (now defined just after the equation), we have changed to \eta which is not used elsewhere in the paper, and defined it (line 265).

o.w. = otherwise. We have simply written "otherwise"

321, 322: Is weight \$\gamma\$ missing here? Is \$c\$ a function in the integral after the first equality and a vector in the second term after the second equality?

\$\gamma\$ is "absorbed" into the definition of the prior covariance (eq 11).

As for the formatting of \$c\$ this was an oversight -- thank you. In the first equality it is a finite element function and we use the capital \$C\$ as previously in the subsection. In the second it is a coefficient vector and we use \overline{c} (cf line 338).

515: The uncertainty in the QoI is not only lower than the uncertainty in the parameters due to the filtering but also due to the choice of QoI. With a different QoI it may be larger even with filtering.

This is a good point -- and we feel it is now addressed by the paragraph which \*now\* follows this one (though, as your correctly point out below, we had forgotten to move to the discussion).

Technical corrections

line 138: \$J\_c\$ -> \$J^c\$

fixed.

176: \$H\$ -> \$\bar{H}\_{mis}\$ ?

We believe this is correct as written. We have added additional equalities to make this clearer.

223: (Section ?)

fixed.

302, 304: \$C^2\$ or \$C\$ here?

\$C^2\$ is what was intended. Eqn (30) is consistent with eq (24). Eq 31 is modified to make clear we are defining C.

385: say something about ..... intervals?

Apologies, we overlooked this leading up to submission -- it is now removed.

413: maybe move ....?

We overlooked this but now it is moved and addresses nicely a comment you make above.

665, 677: missing journals

**Comments from Reviewer 2**

This paper presents a mathematical method and a software tool for quantifying uncertainty in glacier flow model projections. The methods consists of (1) using automatic differentiation to compute the action of the full Hessian of the negative log posterior, (2) computing the smallest few eigenvalue / eigenvector pairs of the Hessian in order to find out which directions in parameter space are most important to sample, and (3) computing the uncertainty in a given output quantity of interest by linearizing the model physics around the most probable state. The authors then test this method on a synthetic test problem, the ISMIP-HOM test case. The authors also address two key shortcomings of how data assimilation in glaciology is usually practiced: accounting for discrete (as opposed to dense) spatial observations and correlation between measurements. Overall, the method is described fairly well albeit with a few points that could use clarifying. I have a few concerns about how generalizable the method is, but in any case the paper and the software are a valuable contribution that I recommend for publication with minor revisions.

**Many thanks for your supportive comments and concise summary.**

Regarding generalisability, we cannot address this point within the context of our manuscript, though since the time of its writing we have been applying the methodology to a more realistic domain and setting. Still, we are glad to hear that even with the idealised experiments here you feel it is a valuable contribution.

The authors make excellent use of low-rank approximations to the Hessian. This trick has appeared in the literature before (although it's not as widely used as it should be). How does their approach compare to that of, say, Petra et al. 2014 or the hIPPYlib code?

In truth the approach is very similar to that of Petra 2014 (which we had neglected to cite, thank you for raising this), which actually follows from their "Part 1" paper, Bui-Thanh et al 2013. The two approaches are very similar and differ subtly, but not profoundly, and essentially do the same thing. We prefer the formulation of Isaac et al 2015, which we follow (but also make explicit some ideas not in their manuscript, such as eq 17). We have already stated that our analysis follows Isaac 2015, but we now state:

**"The following low-rank approximation follows from \cite{Isaac2015} and similar approaches are used in \cite{BuiThanh2013} and \cite{petra2014}"**

I have two real concerns with the approach, although these don't change my overall opinion of this very good paper. First, the method relies on the assumption of quasi-linearity in several places. While the authors check that this assumption wasn't violated for their particular test case, it's difficult to assess whether this would generalize to other problems. By contrast, the stochastic Newton approach in Petra et al. 2014 uses an assumption of quasi-linearity only locally, to bootstrap a more sophisticated Monte Carlo sampling algorithm.

We agree that this approach is limited by the assumption of Gaussianity (ie. quasi-linearity) and we do caveat with the fact this assumption is made, in several places. It is worth noting that Petra 2014's Stochastic Newton MCMC method actually does rely on a low-rank Hessian approximation, which is a central part of the methods discussed in our manuscript as well. While we did not mention Petra 2014 originally, there is a mention of Martin et al 2012, who applied Stochastic Newton MCMC to seismic inversion -- though this was a bit buried in line 422 (of the original manuscript). We now cite Petra 2014 here as well.

The argument we make is that if one wanted to carry out MCMC on a state space this large, they would likely need to use a method such as that of Martin et al 2012 (or Petra et al 2014), which involves (as you mention) a local Hessian based approximation to the posterior pdf -- but even so, these methods are beyond the scope of the current study. To make this more clear we now add text to the discussion making this clear, and note that Stochastic Newton MCMC using our framework could be a future possibility.

Second, the authors make a big deal about using the full Hessian instead of the Gauss-Newton approximation while at the same time relying on quasi-linearity, and yet the Gauss-Newton approximation is works best when the dynamics are almost linear. At the end of the paper, the authors state that they did not use a time-dependent control method

because it's computationally expensive. Would a time-dependent method have been feasible if the authors had instead used the Gauss-Newton approximation? Establishing whether the full Hessian is really necessary is a very important point. Other researchers might want to emulate the techniques described in this paper and yet they might be using other modeling frameworks for which the Gauss-Newton approximation is feasible to implement while the full Hessian is not.

Thank you for this. Based on your feedback a decision was made to implement the Gauss-Newton Hessian within fenics\_ice. We now have an additional section before the Discussion explaining its construction and providing results of a simple test, as well as a short paragraph in the discussion regarding the implications for future work -- but we stress that the results do not necessarily extend to more realistic settings.

**General comments**

45, "gradient-based optimisation": The point of this sentence is to state that computing the MAP estimate doesn't give any information about parametric uncertainty. The fact that you used a gradient-based algorithm concerns more the "how" than the "what"; it isn't really important and you could just cut this phrase entirely. You could have used a derivative-free optimization algorithm -- it's a dreadfully awful idea, but you could do it!

We agree that the MAP point does not tell us anything about uncertainty, we would not completely agree that this is the point we are making -- a point which we feel is addressed in the following paragraph regarding the seemingly contradictory aims of large-scale and UQ. Therefore we feel the efficiency of control methods in finding the MAP point relative to MC methods is in fact relevant. We remove the phrase, now stating

"Although control methods might efficiently provide estimates of parameter fields, they do not provide parametric uncertainty."

57-60: How much better is using the full Hessian than using the Gauss-Newton approximation? This isn't immediately apparent from the text or from the sources you cite here.

**We do not know of another study which addresses this question in the context of UQ. As mentioned, though, we now have a section devoted to this. We of course cannot address this question generally but we can and do for our experiments.**

97: By taking the parameter-to-observation map f to be a map from Rn to Rm, you're assuming a "discretize, then optimize" mindset. It might clarify the points you make about mesh dependence later on if you instead define it as a map from some function space Q to Rm -- the "optimize, then discretize" mindset. See Gunzburger 2002.

We appreciate what you are saying here, but at the same time feel that our approach is very different from the "discretize, then optimize" approach taken by, e.g. the line-by-line differentiation of ocean and climate models by source-to-source transformation (e.g. Kalmikov and Heimbach 2014) -- we mention this because this is the type of approach which "discretize, then optimize" typically refers to. Since tlm\_adjoint differentiates at the level of finite element variational forms, at least in simple cases discretization and algorithmic differentiation commute, in which case the approaches are equivalent.

Regardless, though, we believe our approach in this paper -- to consider functions (and their higher-order derivatives) in terms of coefficient vectors -- greatly simplifies things relative to trying to express relationships as differential forms, which we feel would add complexity to the text and make it more difficult for interested audiences to follow. We are very careful to use distinct notation for finite element coefficient vectors to distinguish from the actual functions, and feel that we have struck a balance between discrete and continuous.

The parameter-to-observation map that you've written down encapsulates both the physics and the measurement process. This is just a suggestion, but it might help the exposition to instead define f as the composition of two maps. First, there's a function g that takes the parameters to the observable fields, like the ice velocity. This function g is basically just "solve the shallow shelf equations". Next, there's a function h that takes the observable fields to the actual observations. When doing the "wrong thing" that you point out later, h is the identity map and the norm is an L^2 norm. When doing the correct that that you have actually implemented, h evaluates the observable fields at a bunch of discrete points and packs these observations into a vector, and the model-data misfit is a discrete sum of squared errors.

Thank you for noting this, as this is a subtle point in the manuscript not made explicit. Introducing another symbol altogether would add complexity to a number of expressions which we feel outweighs the benefits of making this idea clear symbolically. We add a paragraph now in section 4 where Numerical Experiments are described:

"Our parameter-to-observable map \$\vobs{f}\$ is really a composition of two functions: the first finds the solution to the momentum balance (Eq. \ref{eq:weak\_form}) as a finite-element function, and the second interpolates the function to discrete locations. If the misfit cost were to be expressed as the weighted \$L\_2\$ norm of the model-data misfit as in Eq. \ref{eq:control\_mis}, then the interpolation function is replaced by the identity." 135-139: Using a control method does not necessarily imply that you're writing the model-data misfit as a squared L^2 norm, it's just a sinful thing that many glaciologists (including me) have done because it's easier.

We agree with this, and we did not mean to imply that all control methods do this -rather those which are specifically referenced in the introduction do. To avoid ambiguity we now write

**"By contrast with Bayesian methods, the control methods generally used in glaciological data assimilation" .. and cite a subset of those cited in the introduction (to avoid length/repetition)**

It's also worth noting that when you say "mesh dependence", certain readers are immediately going to think of something other than what you describe here. In the PDE-constrained optimization literature, mesh dependence refers to what happens when you use a bad optimization algorithm based on using the vector of coefficients of the derivative obtained from the adjoint method as a descent direction. This can give really obvious mesh imprinting artifacts in the results, especially with higher-order finite element bases. (The minimal right thing to do is to multiply by the inverse of the mass matrix. You mention using the BFGS method later, and taking the H 0 matrix to be the inverse mass matrix works there as well.) This problem is more a question of how you solve a particular optimization problem. The mesh dependence that you're talking about is much more serious -- by neglecting the discreteness of the observations, going to a finer mesh implicitly assumes that you magically have more measurements than you did before. At a higher level, what you've done is tackle the fact that everyone has been solving the wrong problem, irrespective of how they were solving it. Since some readers will immediately associate with the first case, it might be good to either (1) clarify the distinction with a reference to, say, Schwedes et al. 2017, or, (2) if you don't feel like talking about that, use a different phrase besides "mesh dependence".

In fact, by mesh dependence we actually intend the meaning to be with respect to the computational mesh, i.e. how things change (or do not change) when the mesh itself is resolved and finite-element degrees of freedom increase -- which we believe is distinct from the two types of mesh dependence you mention. In the experiment you refer to the observational spacing and location remains the same (we believe this is clear, since Figs 4c and 4d relate to the refinement of the observation array.) We would like to retain "mesh (in)dependence" as it was the same wording used in Isaac et al 2015 (their figure 5), and we now cite this paper where it is mentioned.

140-146: Including the delta term is essentially adding the prior information that you think the mean of the parameters is zero. I don't think this is a good prior in all cases. Are there other ways to get a prior with bounded covariance that don't make this assumption? For example, you could use the Moreau-Yosida regularization of bounds constraints, which

instead assumes that the parameters don't wander too far outside a preset interval but which provides no constraints within that interval.

We agree it is not the only way to impose this prior mean, and agree it is perhaps not the best approach in all cases, but as mentioned in our response to referee 1 we feel that a prior mean of zero is appropriate. The elliptic operator we use for a prior does have strong advantages though such as the ease of inverting which needs to be done many times. We point out this is not the only way to impose this and reference an optimisation study which uses the regularisation method you mention.

260-264: It's easy to get the impression from this paragraph that you're doing time-dependent inversions, which is only dispelled in the conclusion on line 552. You should probably state this earlier in the text.

We now add " In this study, we do not consider initialisations based on time-varying data (i.e. the misfit cost function \$J\_{mis}^c\$ does not depend on time-varying fields), so the continuity function is only involved with finding a Quantity of Interest and propagation of initialisation uncertainty."

270-273: How do you know how many Picard iterations is adequate? Why not use another globalization strategy, like damping / line search or trust regions?

We do not know (or specify) a priori a set number of picard iterations, rather we specify a relative tolerance, which is chosen empirically. We now state this as well as the value used for our study.

274: Why did you use BFGS when you can calculate a Hessian-vector product? Why not Newton-Krylov?

Full Hessian-vector multiplication costs the tangent-linear + second order adjoint, as well as the usual forward + first order adjoint, so as a baseline would be more expensive, and we are not sure that a Newton-Krylov approach would on the whole offer an advantage. While we have not carried out a detailed study of Newton-Krylov methods for optimisation of the cost function, results with a Newton-CG approach were not very promising. Given this, we feel the passage is best kept as-is.

341: The L-curve is fine, but you might want to mention the discrepancy principle or other more statistically-motivated approaches. See Habermann et al. 2013.

We now note that there are alternative approaches to determining the optimal level of regularisation and cite a few references which implement these, including Habermann 2013.

484-490: This is really great, I haven't seen anyone address this issue before.

Thank you, to our knowledge it had not been examined in the context of this problem.

537-538: I don't think the technical hurdles are minor at all because of exactly what you say in the next sentence.

I would remove this statement from the text.

We meant to say that it is relatively trivial to add another "control" field into the fenics\_ice framework and implement Hessian-vector products with respect to this joint space -- not to overcome the issue of equifinality and/or quantify overfitting (in fact this is very difficult). To avoid confusion we modify to

"The version of \texttt{fenics\\_ice} presented in this study is not capable of joint inversions or of Hessian-vector products with multiple parameter fields, however the technical hurdles to implementation are minor."

550-551: It might be worth citing some of the work that Karen Willcox and her group have done on multi-fidelity modeling and UQ.

What we were suggesting was much less sophisticated than a multifidelity approach but we do now add a reference.

**Technical corrections**

Several of the authors' "note to self" comments remain in the manuscript.

Thank you, these were overlooked and now actioned.

484: "isotroptically" -> "isotropically"

Thank you, fixed.
* * *
References

[revised manuscript text omitted]